

# New particle formation in marine atmosphere during seven cruise campaigns

Yujiao Zhu[1], Kai Li[2], Yanjie Shen[1], Yang Gao[1], Xiaohuan Liu[1], Yang Yu[1], Huiwang Gao[1], Xiaohong Yao[1]*

[1]Key Lab of Marine Environmental Science and Ecology, Ministry of Education, Ocean University of China, Qingdao 266100, China
[2]National Marine Environmental Forecasting Center, Beijing, 100081, China

*Correspondence to*: Xiaohong Yao (xhyao@ouc.edu.cn)

**Abstract.** To study the particle number concentration, size distribution and new particle formation (NPF) events in marine atmosphere, we made measurements during six cruise campaigns over the marginal seas of China in 2011-2016 and one campaign from the marginal seas to the Northwest Pacific Ocean (NWPO) in 2014. We observed relatively frequent NPF events in the atmosphere over the marginal seas of China, i.e., 23 out of 126 observational days with the highest occurrence frequency in fall, followed by spring and summer. 22 out of 23 NPF events were analyzed to be associated with the long-range transport of continental pollutants based on 24-hr air mass back trajectories and the pre-existing particle number concentrations largely exceeding the clean marine background, leaving one much weaker NPF event to be likely induced by oceanic precursors alone and supported by multiple independent evidences. Although the long-range transport signal of continental pollutants can be clearly observed in the remote marine atmosphere over the NWPO, NPF events were observed only in 2 days out of 36 days. The nucleation mode particles (<30 nm), however, accounted for as high as 40%±13% of the total particle number concentration during the NWPO cruise campaign, implying that there were many undetected NPF events in the sea-level atmosphere or above.

To better characterize NPF events, we introduced a term, i.e., the net maximum increase in nucleation mode particles number concentration (NMINP) and correlated it with formation rate of new particles (FR). We found a moderately good linear correlation between NMINP and FR at FR ≤8 cm$^{-3}$ s$^{-1}$, but there was no correlation at FR >8 cm$^{-3}$ s$^{-1}$. The possible mechanisms were argued in terms of roles of different vapor precursors. We also found a ceiling existing for the growth of new particles from 10 nm to larger size in most of NPF events. We thereby introduce a term, i.e., the maximum geometric median diameter of new particles ($D_{pgmax}$) and correlate it with the growth rate of new particles (GR). A moderately good linear correlation was also obtained between $D_{pgmax}$ and GR, and only GR larger than 7.9 nm h$^{-1}$ can lead to new particles growing with $D_{pgmax}$ beyond 50 nm. Combining simultaneous measurements of the particle number size distributions and cloud condensation nuclei (CCN) at different super saturations (SS), we indeed observed a clear increase in CCN when the $D_{pg}$ of new particles exceeded 50 nm at SS=0.4%. However, it was not the case for SS=0.2%. Consistent with previous studies in continental atmosphere, our results implied that 50 nm can be used as the threshold for new particles to be activated as CCN in the marine atmosphere. Moreover, the κ decreased from 0.4 to 0.1 during the growth period of new particles, implying that





organics likely overwhelmed the growth of new particles to CCN size. The chemical analysis of nano-MOUDI samples revealed TMA and oxalic acid may play the important role in the growth of new particles.

## 1 Introduction

New particle formation (NPF) events have been widely studied in clean or polluted continental atmospheres since the processes are considered to be an important source of atmospheric particles (Kulmala and Kerminen, 2008; Zhang et al., 2012; Sabaliauskas et al., 2012; Zhu et al., 2014). The grown new particles in the atmosphere can affect Earth's radiation budget by acting as cloud condensation nuclei (CCN) or directly scattering and absorbing solar radiation (Kerminen et al., 2012; Seinfeld and Pandis, 2012). Ocean is the major source of moisture on earth and NPF events in the marine atmosphere are important by considering potential indirect climate impacts. Kazil et al. (2010) reported that the contribution of atmospheric nucleation processes to the total absorbed solar short-wave radiation was -2.18 $W/m^2$ over the oceans, which was seven times larger than that over the continents.

In the clear marine atmosphere, oxidation products of dimethyl sulfide (DMS) and iodine compounds together with reactive amines are generally considered to be the important biogenic precursors of nucleation and growth of newly formed particles (e.g., Charlson et al., 1987; O'Dowd and de Leeuw, 2007; Quinn and Bates, 2011; Sellegri et al, 2016). In the past three decades, many observational and modeling studies have been conducted to investigate the roles of DMS on NPF events in various marine atmospheres and related impacts on the climate, e.g., Charlson et al., 1987; Ayers and Gras, 1991; Hegg et al., 1991; Yu and Luo, 2010; Chang et al., 2011; Lana, et al., 2012. The responses of grown new particles and clouds to marine biogenic products are much more complex than those derived from DMS (Quinn and Bates, 2011). NPF events induced by iodine species were frequently observed at coastal locations, e.g., most notably at Mace Head on the west coast of Ireland, in presence of enhanced biological emissions during low-tide, ocean upwelling or the sea-ice melting conditions (O'Dowd et al., 2002 a,b; Wen et al., 2006; O'Dowd and de Leeuw, 2007; Ehn et al., 2010; Huang et al., 2010; McFiggans, et al., 2010; Allan et al., 2015; Sellegri et al., 2016). Moreover, amines were reported to enhance $H_2SO_4$-$H_2O$ nucleation and promote the growth of newly formed particles (Ge et al., 2011; Zhang et al., 2012; Yu et al., 2016; Olenius et al., 2017). The observation in the eastern Pacific Ocean revealed that an increase in gaseous precursors of amines and DMS affected the particulate chemical composition and cloud properties (Sorooshian et al., 2009). Lab studies showed that reactions of methanesulfonic acid (MSA) and amines can generate new particles, even in the absence of sulfuric acid (Chen et al., 2015 a,b; Arquero et al., 2017).

The moderate cloud water vapor super saturations (SS) are around 0.2% in the marine boundary layer (MBL), needing the minimum size of 70-80 nm particles in diameter to be activated into CCN (Hoppel et al., 1996; Petters and Kreidenweis, 2007; Seinfeld and Pandies, 2012). In the atmosphere over the eastern Mediterranean in summer, newly formed particles can grow to approximately 100 nm in a few hours because of the clean air masses abundant in $H_2SO_4$ vapor from high altitudes, and the number concentration of CCN substantially increased (Kalkavouras et al., 2017). In some iodine-related NPF events, newly formed particles can also grow up to approximately 100 nm in a few hours, and significantly contributed to the CCN





concentration (O'Dowd et al., 2002b). However, no obvious growth of new particles was observed in a large number of the iodine-related NPF events and were classified as local NPF events (Wen et al., 2006; Vana et al., 2008; Ehn et al., 2010). Some previous cruise measurements in open oceans showed that NPF events were rarely observed and did not contribute significantly to the CCN concentration (Covert et al., 1996; Ueda et al., 2016).

Unlike the clear marine atmosphere, oxidation products of air pollutants such as $SO_2$, NOx, and anthropogenic volatile organic compounds might be also involved in NPF events in polluted marine atmosphere (Pikridas, et al., 2012; Liu et al., 2014; Peng et al., 2014; Guo et al., 2016). Theoretically, the continental air pollutants and marine biogenic precursors may interact with each other and affect the formation and growth of new particles. It is poorly understood what is the difference among NPF events in the marine atmosphere driven by the long-range transported continental pollutants, ocean-derived

biogenic precursors, or a combination of both.

In this study, we investigated NPF events during seven cruise campaigns covering three seasons, including spring, summer and fall over the marginal seas of China (including the Bohai Sea (BS), North Yellow Sea (NYS), South Yellow Sea (SYS) and East China Sea (ECS)) to the Northwest Pacific Ocean (NWPO). When the summer Asian monsoon is prevailing, the wind direction is mostly from the sea to the land in the daytime and increases the probability to successfully observe NPF

events associated with the ocean-derived precursors. Spring and fall are the transition periods between the summer and the winter Asian monsoon. NPF events during the campaigns could be related to ocean-emitted and/or continent-transported precursors and will be used for comparison. In addition, a measurement at a coastal site was performed simultaneously with marine measurements during three campaigns to identify regional NPF events and reveal similarity and difference between them in marine and coastal atmospheres. All NPF events in marine atmosphere and supporting observations in the coastal

atmosphere were analyzed in terms of occurrence frequency, formation rates and growth rates of new particles, the maximum increase of new particles in number concentration and the maximum size of grown new particles, the new particle survival probability to CCN sizes, possible chemicals driven the new particles growth, etc.

## 2 Experimental design

### 2.1 Cruise description and sampling methods

Measurements were made in six cruise campaigns across the marginal seas of China including the BS, the NYS, the SYS and the ECS (Fig. 1a-c) on 16 October - 5 November in 2011, 2-20 November in 2012, 5-25 November in 2013, 28 April - 19 May in 2014, 18 August - 5 September in 2015, and 28 June - 20 July in 2016, respectively. The level of air pollutants indicated by the $NO_2$ column densities in the research sea areas showed the clear seasonal variation (Fig. S1). In November 2012, the $NO_2$ column densities were higher in the eastern mainland of China due to the house-heating, and subsequently affect the

downwind marine atmospheres through the continent outflow driven by the Asian monsoon (Fig. S1a). In May 2014, the atmosphere in the marginal seas also received the continental pollutants to a small extent (Fig. S1b). When the summer Asian monsoon prevailing in August, the clean atmosphere generally occurred over the marginal seas (Fig. S1c). On 18 March - 22



April 2014, we made measurements from the marginal seas to the NWPO where the $NO_2$ column densities showed clean atmosphere in particular over the NWPO (Fig. S1d).

During each cruise campaign, a Fast Mobility Particle Sizer (FMPS, TSI Model 3091) was set up on the sixth floor of *Dongfanghong 2* approximately 15 m above the sea level. The FMPS was downstream of a dryer and continuously measured

particles number concentrations ranged from 5.6 nm to 560 nm in 32 channels in 1-s time resolution. The FMPS was used because its high time resolution can allow successfully isolating the interference associated with ship-self emissions and identifying the growth of newly formed particles from the mixing of newly formed particles and pre-existing particles (Liu et al., 2014; Man et al., 2015). However, FMPS suffers from the weakness, i.e., it was reported to underestimate particle size respect to the Scanning Mobility Particle Sizer (SMPS) and HR-ToF-AMS (Lee et al. 2013). In this study, the FMPS data were

corrected based on a Condensation Particle Counter (CPC, TSI Model 3775) using the method developed by Zimmerman et al. (2015). A Cloud Condensation Nuclei Counter (CCNC, DMT) was set up nearby the FMPS during the four cruise campaigns in 2014-2016. The CCNC was operated at a total flow rate of 0.45 L min$^{-1}$ and set five different SS of 0.2%, 0.4%, 0.6%, 0.8% and 1.0%. A 14-stage nano Micro-Orifice Uniform-Deposit Impactor (nano-MOUDI, Models 122-NR) equipped with Teflon filters in the upper 11 stages and Zefluor filters in the remaining three stages was set up nearby the FMPS during

six campaigns in the year from 2012 to 2016. The 50% cutoff points for particle aerodynamic diameters were 18, 10, 5.6, 3.2, 1.8, 1.0, 0.56, 0.32, 0.18, 0.1, 0.056, 0.032, 0.018 and 0.010 μm. Alternatively, when the nano-MOUDI was out of service, an 11-stage MOUDI (Models 110-II™) with the cutoff sizes from 18 μm to 0.056 μm was used instead. The air pump of MOUDI or nano-MOUDI was switched on only when the ship was sailing, and the sampling duration was approximately 10-12 hours per day, in order to accumulate sufficient mass for chemical analysis. The inorganic ions, aminium ions and dicarboxylic acids

were detected, and the chemical analysis method was detailed in Text S1.

The coastal measurements were conducted in the campus of Ocean University of China (OUC, 36°09′37″N, 120°29′44″E, red circle in Fig. 1) simultaneously with three cruise campaigns on 2-20 November in 2012, 5-25 November in 2013 and 18 August to 5 September in 2015. The sampling site at OUC was approximately 7 km away from the nearest coastline of the South Yellow Sea. A Nanoscan Scanning Mobility Particle Sizer Spectrometer nanoparticle sizer (nano-SMPS, TSI, 3910)

was placed on the fifth floor of an academic building (~ 15 m above the ground level), and continuously measured number concentrations of particles ranged from 10 nm to 420 nm in 13 channels in 1-minute time resolution. The two particle sizers, i.e., FMPS and nano-SMPS, were operated side-by-side for comparing in 14-25 October 2012. The determination coefficients ($R^2$) of particle number concentrations between them were in the range of 0.7-0.9. Fig. S2 showed comparison results during four NPF days. The number concentration of nucleation mode particles ($N_{<30\ nm}$) measurement by FMPS was higher than nano-

SMPS, especially during the initial NPF time, further leading to the calculated new particles formation rate measured by FMPS was 0.2-0.8 fold larger than that of nano-SMPS (Table S1). The difference could also be due to the different size detection limits, i.e., 5.6 nm for FMPS and 10 nm for nano-SMPS. When the particles with diameters smaller than 10 nm was excluded for calculation, the particle number concentrations between 10-30 nm measured by FMPS matched well with those measured by nano-SMPS. The new particle formation rates calculated from data measured between the two instruments had a difference



within 20% (Fig. S2 and Table S1). In this study, particles smaller than 10 nm were included to calculate the new particle formation rate for the data measured by the FMPS, if not specified. In addition, the two fitted geometric median diameter of new particles ($D_{pg}$) showed the good correlations when newly formed particles exhibited the obvious growth. The growth rates calculated from the data measured between the two instruments had a difference within 20%.

## 2.2 Definition of NPF events and calculation methods

NPF events are identified on basis of the criteria proposed by Dal Maso et al. (2005), Hirsikko et al. (2007) and Kulmala et al. (2012), i.e., 1) a new mode of particles, i.e., nucleation mode < 30 nm, must be observed, 2) the new mode must prevail over a time span, 3) the new mode must grow to be observable. The three criteria allow distinguishing the NPF events from ship emissions or regional air pollutant plumes.

The new particle formation rate (FR), growth rate (GR) and condensation sink (CS) were calculated, and the methods were detailed in Text S2. The net maximum increase in nucleation mode particles number concentration (NMINP) was defined as $N_{<30\,nm}$ at the time reaching the maximal value minus $N_{<30\,nm}$ at the time immediately before the apparent NPF initiated (Zhu et al., 2017). The CCN activation ratio (AR), i.e., the number fraction of atmospheric aerosol particles (condensation nuclei, CN) activated to CCN at a given SS, is estimated by the equation as below:

$$AR\ (SS) = N_{CCN}/N_{CN}$$

where $N_{CCN}$ was the number concentration of CCN measured by CCNC, and $N_{CN}$ was the total particle number concentration measured by the FMPS.

The mass concentrations of nanoparticles were calculated using the corrected FMPS data. The electrical mobility diameter was converted to the aerodynamic diameter following the equation of Khlystov et al. (2004), and the density of the particles was assumed to be 1.5 g cm$^{-3}$. Air mass back trajectories were calculated using HYbrid Single-Particle Lagrangian Integrated Trajectory (HYSPLIT) model with REANALYSIS meteorological data. The time indicated in this paper are local time, i.e., UTC+8 over the marginal seas of China, UTC+10 on the locations of two NPF days over NWPO.

## 3 Results

### 3.1 Occurrence of NPF events in marine atmosphere

During the seven cruise campaigns, NPF events were observed in 25 days out of the total of 162 observational days (Tables 1 and A1, Fig. A1, Liu et al., 2014). All NPF events were initially observed at 07:00-12:00 in the morning on sunny days. In the atmosphere over marginal seas of China, the occurrence frequency of NPF event was the highest in fall (28%), followed by spring (14%) and summer (7%). On the contrary, spring is widely documented as the season with the highest occurrence frequency for NPF events in the continental boundary layer globally (Kulmala et al., 2004; Zhu et al., 2014; Wang et al., 2017; Nieminen et al., 2018). Different from these high spring continental NPF events, the spring over the marginal seas was the season with most frequent occurrence of sea fog likely resulting from the slower warming of sea surface relative to





the surrounding air (Roach et al., 1995; Cho et al., 2000), leading to high RH in the MBL and subsequent prevention of NPF occurrence. During the summer campaign, NPF events were rarely observed. Compared to the marginal seas of China, the cruise over NWPO during the spring of 2014 showed much fewer NPF events occurrence, i.e., 2 out of 36 days.

As explained in the Section 2.1, simultaneous measurements at the coastal site (OUC) were conducted during three-cruise periods with a total of 59 days in 2012, 2013 and 2015. NPF events were observed in 16 sunny days out of the total 59 days (right panels in Fig. A1). Regional NPF events observed both in the marine and coastal atmospheres included five days during the fall campaign in 2012, six days during the fall campaign in 2013, but only one day during the summer campaign in 2015. On basis of the distance between OUC and ship location at that time, we roughly estimated the spatial scale for regional NPF events at least 47 km - 475 km. In each regional NPF event, NPF started 15 minutes - 3 hours earlier in the marine atmosphere than the OUC site. Theoretically, concentrations of precursor vapors and condensational sinks were two important factors to determine the occurrence of NPF events (Kulmala et al., 2004; Kulmala and Kerminen, 2008). The delays in the coastal atmosphere in some cases were identified very likely due to stronger condensational sinks, e.g., on 17 November 2012, and 7 November 2013 (Fig. A1h, i, l, m). Many other factors which can affect concentrations of precursor vapors might also contribute to the delay. Noted that the occurrence frequencies of NPF events during the two fall marine campaigns were similar to those in the coastal atmosphere at OUC site, but the frequency during the summer marine campaign was much lower. The marine atmosphere is quite clear in summer due to the summer Asian monsoon.

## 3.2 Particle number concentrations and size distributions in presence of NPF events against the background atmosphere

To compare particle number concentrations and size distributions in presence of NPF events with those in the background atmosphere, we first classify the observational data into two categories. Category 1 represents background particle signals during the periods with neither NPF events nor ship-self emissions. Category 2 represents atmospheric particle signals measured in presence of NPF events, with the ship-self emission signals to be removed. For each category, the average particle number concentration spectra were derived for the marginal seas of China, NWPO and OUC.

The category 1 was first discussed regarding the marginal seas of China and NWPO. The averaged total particle number concentration was $0.8\pm0.5\times10^4$ particles cm$^{-3}$ in the atmosphere over the marginal seas of China (dashed black lines in Fig. 2). The Aitken mode (30-50 nm) and accumulation mode (50-200 nm) were usually overlapped at the size range of 30-200 nm with a minor nucleation mode at the size below 30 nm. The contribution of the Aitken mode particles to the total number concentration relative to that of accumulation mode varied largely in different campaigns (not shown). Compared to the marginal seas of China, the total particle number concentrations decreased to $0.3\pm0.1\times10^4$ particles cm$^{-3}$ over the NWPO, mainly because of a large drop of number concentrations of particles with diameter larger than 20 nm. The decrease is likely due to the combined contribution from dry and wet deposition, cloud processing and coagulation, etc., during the long-range transport of atmospheric particles from the marginal seas or the continents to open oceans (Hoppel et al., 1986; Guo et al., 2016; Luo et al., 2018). The $N_{<30 nm}$ measured over the NWPO ($0.12\times10^4$ particles cm$^{-3}$) was relatively comparable to, albeit



slightly lower than, the marginal seas ($0.17\times10^4$ particles cm$^{-3}$). However, the percentage contribution of nucleation mode particles to the total number concentration over the NWPO (40%) was twice as large as the marginal seas (20%), implying the important role of nucleation mode particles as a source of atmospheric particles over the NWPO. The observed high percentage of N$_{<30\,nm}$ over NWPO was primarily a result of either direct generation in the marine boundary layer or downward transport from high altitude such as free troposphere (Clarke et al., 1998; Kulmala and Kerminen, 2008; Meng et al., 2015; Ueda et al., 2016).

Compared to Category 1, NPF events greatly enhance the total particle number concentrations (Fig. 2, solid lines) in Category 2 over three regions including NWPO, marginal seas of China and OUC. Over the NWPO, the concentration increase (solid read line, Fig. 2) was limited to particles with the diameter less than 20 nm, whereas the concentration increase seemingly extends to particles with the diameter until 60 nm over the marginal seas (solid black line, Fig. 2), possibly because of different growth pathways of newly formed particles. The N$_{<30\,nm}$ of $0.9\pm0.6\times10^4$ particles cm$^{-3}$ over NWPO was higher than previous studies in open seas (Covert et al., 1996; Ueda et al., 2016), and account for 60% of the total particle number concentration. The increase likely induced by the long-range transport of air pollutants from the continents, inferred from the doubled number concentrations of accumulation mode particles in Category 2 relative to Category 1. Over the marginal seas of China, the average total particle number concentration was $1.6\pm1.1\times10^4$ particles cm$^{-3}$, albeit slightly smaller than that at OUC ($2.5\pm0.8\times10^4$ particles cm$^{-3}$), showing comparable net increase, referred to as the arithmetic difference between Category 2 and Category 1. In addition, the average particle number size distributions over the marginal seas of China showed seasonal differences, i.e., a sharp peak occurring in the particle sizes less than 20 nm in spring, a broad peak occurring in the particle size range of 7-50 nm in fall, and an even broader peak in the size range from 10 nm to 70 nm in summer (Fig. S3). The results were caused by varying growing ceilings, which was clearly shown at OUC as well, of new particles growing to the maximum sizes observed during the events in three seasons. In fact, the existence of the ceiling is the common phenomenon during the NPF events occurring in various urban or coastal atmospheres, highlighted by Zhu et al. (2014, 2017) and Man et al. (2015).

**3.3 FR, GR, NMINP and D$_{pgmax}$ in marine atmosphere**

A total of four metrics were applied to evaluate NPF events including two commonly used parameters (FR and GR) and two other parameters, with one (NMINP) developed in our previous study (Zhu et al., 2017) and the other (maximum D$_{pg}$, D$_{pgmax}$) newly developed in this study.

In the atmosphere over marginal seas of China, large variations in FRs and GRs were observed during three seasons (Table 1 and A1), with summer larger than spring and fall. For example, among the total of three NPF events in summer, the larger FRs and GRs were observed on two days, i.e., 18.5 cm$^{-3}$ s$^{-1}$ and 9.6 nm h$^{-1}$ on 27 August 2015, and 12.7 cm$^{-3}$ s$^{-1}$ and 10.6 nm h$^{-1}$ on 4 September 2015, leaving the third one a much smaller value (0.3 cm$^{-3}$ s$^{-1}$ and 1.7 nm h$^{-1}$) on 30 August 2015 likely induced by precursors from marine sources. The high FRs and GRs in summer was reported to be induced by enhanced photochemical reactions resulting from strong solar radiation (Kulmala et al., 2004; Wang et al., 2017). Regarding the two observed NPF events over NWPO, large GRs differences were found, 26.3 nm h$^{-1}$ on 8 April 2014 and 3.6 nm h$^{-1}$ on 13 April



2014. The high GR on 8 April 2014 was concurrent with high FR (11.8 cm$^{-3}$ s$^{-1}$), unfortunately, no FR was available on the low GR day (13 April 2014) due to the intermittent nucleation occurrence. The GR and FR observed here much higher than previous studies over the subtropical Pacific area (Ueda et al., 2016). We will return the NPF events in Section 4.2. Together considering both the marginal seas and NWPO, the FR and GR did not show significant negative correlation with CS.

5       Since the coastal experiments at OUC were designed to simultaneously measure particles with cruises over marginal seas, we therefore compare the NPF events between OUC and marginal seas during the three campaigns. Note that only atmospheric particles with diameter larger than 10 nm in both OUC and marginal seas were used to compare FRs and GRs in this paragraph, because of the detection limit of nano-SMPS. The FRs over the marginal seas (5.5±5.6 cm$^{-3}$ s$^{-1}$) displays larger mean condition and variations in relative to OUC (2.2±0.7 cm$^{-3}$ s$^{-1}$), whereas the comparable GRs were observed at OUC (6.1±4.8 nm h$^{-1}$) and

the marginal seas (4.7±4.7 nm h$^{-1}$). Compared to OUC, the larger FRs over the marginal seas could be due to 1) the lower number concentration of pre-existing particles therein and the associated smaller CS; 2) higher loadings of precursors favorable for the formation of new particles (Hu et al., 2015; Yu et al., 2016). These precursors such as sulfuric acid, amines and extremely low-volatility organic vapors were reported to dominate the formation and initial growth of new particles (Paasonen et al., 2010; Ehn et al., 2014). In contrast, the different GR between the marginal seas and OUC could be related to the

availableness of semi-volatile precursors which promote the growth of larger new particles (Yao et al., 2010; Riipinen, et al., 2012).

        In terms of NMINP, it varied from 438 cm$^{-3}$ to 3.6×10$^{4}$ cm$^{-3}$ in the marine atmosphere, but no obviously seasonal variations were observed as was discussed previously regarding FR and GR (Table A1). Over the marginal seas and NWPO, the NMINP and FR showed a moderately good linear correlation when the FRs were smaller than 8 cm$^{-3}$ s$^{-1}$, with r=0.83 and

p<0.01 (Fig. 3a), but no correlation existed with FRs exceeding 8 cm$^{-3}$ s$^{-1}$. The differences in the correlation may be largely explained by the theory that NMINP was determined by the consumed amount of H$_2$SO$_4$ vapor, whereas FR was controlled by the amount of both H$_2$SO$_4$ vapor and organic vapor, i.e., FR= $k_{NucOrg}$[H$_2$SO$_4$]$^{m}$[NucOrg]$^{n}$ with $k_{NucOrg}$ as constant and m and n as integers (Zhang et al., 2012; Zhu et al., 2017). For example, in Scenario 1 in presence of high correlation, the H$_2$SO$_4$ vapor should be relatively sufficient while the organic vapors are the limiting factor of the FR. In this situation, the available organic

vapor determines FR, which affects the H$_2$SO$_4$ vapor used for nucleation, and the consumed H$_2$SO$_4$ vapor for nucleation determines NMINP. Therefore, the correlation between NMINP and FR is established. In Scenario 2 absent of the correlation, the amount of the organic vapors involved in nucleation and initial growth should be relatively sufficient relative to H$_2$SO$_4$ vapor, leading to larger FR. However, the NMINP was always determined by the consumed H$_2$SO$_4$ vapor for nucleation. Thus, it is not surprising that no correlation between NMINP and FR was observed in Scenario 2. The FRs threshold of 8 cm$^{-3}$ s$^{-1}$

and the slope of 3.9×10$^{3}$ were consistent with our previous study in which more observations in the polluted continental atmospheres were included (Zhu et al., 2017). However, due to the limited data, no significant correlation between NMINP and FR were obtained at OUC (Fig. S4a).

        Since the ceiling of the growing new particle sizes existed, we defined the D$_{pgmax}$ to represent the maximum sizes of growing new particles. Among the total of 25 NPF events over marginal seas and NWPO, the D$_{pgmax}$ was smaller than 50 nm



in the 24 NPF days except one day of 77 nm. As presented later, no clear increase in $N_{CCN}$ can be identified when the $D_{pg}$ increases up to 77 nm at SS=0.2%. However, when the SS set to 0.4%, the $N_{CCN}$ clearly increases with the $D_{pg}$ of newly formed particles increasing beyond 50 nm. Thus, we considered 50 nm as the CCN size under moderately high SS. A moderately linear correlation between $D_{pgmax}$ and GR in the daytime was obtained by excluding an outlier of 26 nm h$^{-1}$ (Fig. 3b). Based on
the obtained slope, newly formed particles can grow to the size larger than 50 nm only when the GR exceeded 7.9 nm h$^{-1}$ in the daytime. At OUC site, a smaller GR (4.8 nm h$^{-1}$) was required for new particles to grow to the CCN size, probably because of the longer survival time of new particles (Fig. S4b). The growth of new particles to CCN size will be presented in section 4.3.

### 3.4 Growth patterns of newly formed particles

Two growth patterns (denoted to Class-I and Class-II) of newly formed particles were observed in the atmosphere over the marginal seas, NWPO and at OUC site. Class-I NPF events were characterized by a typical "banana-shaped" growth when the $D_{pg}$ increased from approximately 10 nm to 20-100 nm in 0.5-18 h, which occurred in sixteen days over the marginal seas and thirteen days at OUC site (e.g., Fig. 4a,b). For Class-II NPF events, either no growth or extremely low growth of newly formed particles were observed. The $D_{pg}$ remained invariant or slightly increased, but it was less than 20 nm until the number
concentration of newly formed particles dropped to negligible levels (e.g., Fig. 4c, d). Compared to Class-I NPF events, fewer Class-II events were observed in both marginal seas of China (seven days) and OUC (three days). Over NWPO, the two NPF events were classified as Class-II, due to the small $D_{pgmax}$ of 14 nm, although the large GRs were calculated.

     Traditionally, Class-I NPF events usually occurred in regional scales over a variety of atmospheres such as forested areas, rural and urban areas, and even in heavily polluted regions (Kulmala and Kerminen, 2008), whereas Class-II NPF events were
mostly reported as a local phenomenon by a few studies over coastal areas (Wen et al., 2006; Vana et al., 2008; Ehn et al., 2010). In our previous study (Zhu et al., 2017), Class-II NPF events were observed in urban atmosphere and further defined as a regional phenomenon instead of a local event. Thus one of our objectives to design the parallel experiment over marginal seas and OUC is to confirm whether the Class-II NPF events can be regional occurred in MBL and to be observed at both sites. The result is that a total of seven Class-II NPF events were found over the marginal seas (Fig. A1c, h, v, z, aa, ac, ah), and
three NPF events can be concurrently observed over both marginal seas and OUC (Fig. A1d, i, w), regarding the other four events, except during one event no NPF event was observed at OUC, no measurement was available during the other three events. Moreover, the duration of all the seven events lasts an hour or longer. Thus, the Class-II NPF events over the marginal seas in this study seemed to be occurred in a regional scale.

     To delve into more details regarding the three concurrent NPF between marginal seas and OUC, during two of them at
OUC was Class-II (i.e., 12 and 17 November 2012, Fig. A1d, i) as well, however, the third one at OUC belong to Class I (19 November 2013, Fig. A1w). The distances between the two locations were 246 km, 243 km and 467 km during these three events, and the air mass back trajectories indicated that the two locations were affected by large-scale regional air masses (Fig. S5b, d, l).



Compared with Class-I NPF events, no apparent growth or the extremely low growth of new particles during Class-II NPF events implied that semi-volatile species thermodynamically can not support the particle growth, but semi-volatile species likely dominate the contribution to the growth of >10 nm particles in Class-I. On 19 November 2013, a Class-II event observed over the marginal sea concurrently with a Class-I event at OUC, implied that the vapor pressure of semi-volatile species in the

marine atmosphere were less than the required value for gas-particle condensation, but the particle growth in the coastal atmosphere implied the vapor pressure of the species exceeding the required value.

## 4 Case studies and discussion

### 4.1 NPF events induced by ocean-derived precursors over the marginal seas of China

During the spring or fall cruises over the marginal seas in 2011-2014, the 24-hr back trajectories showed that air masses

passed through areas with large emissions of air pollutants when NPF events occurred (Gao et al., 2011) (Fig. 5a and S5). In contrast, during the summer cruises of 2015, the three observed NPF events was accompanied by air masses transport mainly from the marine atmosphere based on back trajectories (red lines in Fig. 5a, Fig. S5r, t, w). For the three summer events, two of them were from the air masses passed through the Shandong Peninsula more or less, whereas the third one on 30 August 2015, a weak NPF occurred when the air mass almost completely came from the marine atmosphere.

To delve into the characteristics and evidence of oceanic precursors related NPF event on 30 August 2015, the transport pathway on that day was first zoomed in and showed in Fig. 6a. As is illustrated in Fig. 6b, the NPF event started to be observed at 09:40, with meteorological conditions such as ambient temperature of 26°C, high RH at 74%, and low wind speed at 1.5 m s$^{-1}$ (not shown). This weak NPF event was featured by the low particle number concentration and small $D_{pg}$. For example, the number concentration of nucleation mode particles increased from 600 cm$^{-3}$ to 2000 cm$^{-3}$ in the initial one hour, and the

NMINP was 5-20 times lower than all the other NPF events over the marginal seas. The subsequent FR of 0.3 cm$^{-3}$ s$^{-1}$ was the minimum in this study. No apparent growth of new particles in the initial thirty minutes, and then the $D_{pg}$ fluctuate between 10 nm and 13 nm in the next hour, followed by $D_{pg}$ increase from 13 nm to 18 nm in another 3 hours with the growth rate of 1.7 nm h$^{-1}$. To further support the marine source of this NPF event, one set of nano-MOUDI samples was collected during the period from 7:40 to 23:00 on that day. The mass concentrations of particulate dimethylaminium (DMA$^+$) and trimethylaminium

(TMA$^+$) were elevated in the small size range of 10-56 nm and might contribute to the formation and growth of new particles to some extent (Fig. 6c). The observed high DMA$^+$ and TMA$^+$ were very likely derived from oceanic sources as reported by Hu et al. (2015) and Yu et al. (2016). Moreover, the simultaneously in-situ measured subsurface chlorophyll maximum (SCM) of seawater as high as 3 mg m$^{-3}$ during the NPF period also suggested strong biological activity in the sea zone. Overall, the NPF induced by oceanic precursors alone appeared to be much weaker than those by the precursors derived from continents

or the combination of both. For example, in the marine atmosphere, the observed oceanic particulate MSA was about two orders of magnitude smaller than particulate sulfate (Zhang et al., 2014).



## 4.2 NPF events over the NWPO

In the atmosphere over the NWPO, only two NPF events were observed on 8 and 13 April 2014 (Fig. 7). On 8 April 2014, the NPF event started to be observed at 7:43 and lasted for 17 minutes (Fig. 7a, UTC+10). The total particles number concentration increased from $0.6\times10^4$ cm$^{-3}$ to a maximum value of $2.7\times10^4$ cm$^{-3}$ during the NPF event. The NMINP of $1.2\times10^4$ cm$^{-3}$ ranking at the moderate level of those values over the marginal seas of China. The initial $D_{pg}$ was about 8 nm, and grew to 14 nm till the signal of new particles disappeared. The calculated FR and GR were 11.8 cm$^{-3}$ s$^{-1}$ and 26.3 nm h$^{-1}$, higher than those of NPF events over the marginal seas of China. In the marine atmosphere of open oceans, the NPF events may occur aloft instead of sea level atmosphere (Wiedensohler et al., 1996; Clarke et al., 1998). The high number concentration of new particles over NWPO was likely a result from downward effect of long-range upper-level continental transport, with the following two indirect evidences. First, the number concentrations of atmospheric particles larger than 30 nm reached 5000-6000 cm$^{-3}$. The values were one order of magnitude larger than the clean marine background and were approximately two times larger than the corresponding average values in non-NPF days (Category 1 discussed in section 3.2). In addition, the 24-hr air mass back trajectory indicated the air pollutant transport from the south of Japan at the altitude of 2000 km (Fig. 7c), together with the long transport distance, the increased particle number concentrations over that far remote oceanic zone were more likely linked with the transport of air pollutants aloft and then mixed down.

In order to elucidate which chemicals may involve in the NPF event, one set of MOUDI samples collected during the period from 11:12 to 23:33 on that day further analyzed. Noted that the MOUDI was used for sampling during this cruise, because of the outbreak of the nano-MOUDI's pump. The mass concentration of sulfate and oxalate in particles less than 10 μm was 2.3 μg m$^{-3}$ and 0.12 μg m$^{-3}$ (derived from Fig. 7d, e), respectively, and higher than other non-NPF days over the NWPO, which has been discussed in detail by our previous study (Guo et al., 2016). The elevated concentration of sulfate and oxalate were mainly related to the enhanced anthropogenic precursors (Fig. 7d, e). In addition, the marine derived precursors may also involve in the NPF events, supported by the moderate high level of TMA$^+$ and DMA$^+$ mass concentration in particles less than 10 μm (0.08 μg m$^{-3}$ and 0.03 μg m$^{-3}$), the substantially elevated TMA$^+$ in 56-320 nm particles (red cross in Fig. 7d) as well as the relatively high MODIS-derived chlorophyll-a (Fig. S6). As was mentioned earlier, due to the outbreak of the nano-MOUDI, no chemical components in particles smaller than 56 nm were observed, thus the contributions of TMA in the formation and growth of new particles need to be further investigated.

Compared to the event above on 8 April, the other one on 13 April showed similar features and sources, albeit with slightly longer duration starting from 07:50 to approximately 08:50 (Fig. 7b). The new particles signal was intermittently observed and the FR was difficult to calculate. The total particle number concentrations increased from $0.3\times10^4$ cm$^{-3}$ to the maximum of $2.6\times10^4$ cm$^{-3}$ during the NPF event, and the NMINP was $1.4\times10^4$ cm$^{-3}$. The $D_{pg}$ increased from 8 nm to 14 nm in one hour, and the estimated GR was 3.6 nm h$^{-1}$. The mass concentration of sulfate and oxalate in particles less than 10 μm was 0.7 μg m$^{-3}$ and 0.05 μg m$^{-3}$, indicating a smaller anthropogenic impact compared to the other event on 8 April (Fig. 7f, g). The





mass concentration of TMA$^+$ in particles less than 10 μm were 0.05 μg m$^{-3}$ with obviously elevated concentration only in particles of 56-100 nm, indicating the marine derived precursors may involve in the NPF event.

### 4.3 The growth of new particles to CCN size and the maximum survival probability

Limited by the availability of CCN measurements in marine atmosphere, there was only one NPF event on 4 September

2015 over the BS when the grown new particles were clearly identified to contribute to CCN at SS=0.4% (Fig. 8a, b). The event was observed at the initial high ambient RH of 73%. The total particle number concentration increased from $1.0×10^4$ cm$^{-3}$ s$^{-1}$ to the maximum value of $4.5×10^4$ cm$^{-3}$ s$^{-1}$ at 11:12, followed by a decrease to $2-3×10^4$ cm$^{-3}$ s$^{-1}$ in the next three hours (solid line in Fig. 8c). At SS of 0.4%, the $D_{pg}$ increased from 19 nm to 50 nm during 10:40-13:10 (black circles in Fig. 8a) with AR fluctuating at 0.1-0.2 (Fig. 8c). After 13:10, the $D_{pg}$ increased from 50 nm to 77 nm with increasing AR from ~0.2 to ~0.4,

suggesting that 50 nm was the threshold for the grown new particles activating as CCN at SS=0.4% (Dusek et al., 2006; Petters and Kreidenweis, 2007). However, the contribution of grown new particles to $N_{CCN}$ at SS=0.2% was unidentified. Assume that the grown new particles exist a log-normal particle number size distribution, and the grown new particles mode was mainly within particles diameter ($D_p$) subject of μ±3σ nm (μ was the mean value and σ was the standard deviation in the log-normal distribution). We then calculated $N_{>Dp}$ with $D_p$ varying from 50 nm to the maximum value of μ+3σ (about 180 nm in this case),

and for each $D_p$, the values $N_{>Dp}$ was correlated with $N_{CCN}$ measured at SS of 0.4%, and the desirable $D_p$, which will be used to calculate effective hygroscopicity parameters (κ) next, needs to satisfy a linear regression slope close to 1 with relatively high correlation. In this case, we found $D_p$ of 60 nm meets this criteria during the initial NPF from 10:50 to 11:48. The subsequently calculated κ was 0.40, smaller than that of marine atmospheric aerosols (~0.7), but close to that of continental atmospheric aerosols (~0.3) (Poschl et al., 2009; Rose et al., 2010), indicating the continental influence on this NPF event. The

same method was applied to the new particles growing period from 12:40 to 15:02, and the $D_p$ of 92.5 was identified to yield best result (i.e., slope of 1.03 and r=0.9), with estimated κ as low as 0.1, suggesting the condensation of organic vapors on the growing new particles (Yu, 2011; Riipinen, et al., 2012).

To further evaluate the contribution of new particles to CCN, the maximum survival probability of new particles to CCN sizes (SP) was estimated as $N_{50\ nm-(μ+3σ)\ nm}/N_{<30\ nm}$, where the $N_{50\ nm-(μ+3σ)\ nm}$ and $N_{<30\ nm}$ refers to as the maximum value during

the event. In this case (4 September 2015), the SP was calculated to be 0.83, implying majority of newly formed particles can grow to activate as CCN at SS=0.4%. Following this approach, the estimated SP ranged from 0 to 0.29 with the averaged value of 0.05 during the remaining 24 NPF events over the extended areas including the marginal seas and NWPO. Based on this low SP value, there were low probability for the grown new particles to activate as CCN. Moreover, there were three days in the coastal atmosphere when the $D_{pg}$ grew to 70-100 nm (Fig.A1o, q, u). Unfortunately, no simultaneous measurement of CCN

was available.



## 4.4 The roles of amines and oxalic acid in growing new particles

The oxidized organic compounds were widely identified to promote the growth of new particles, but most of these studies are restricted at the level of volatility and functional groups instead of the specific organic compounds (e.g., Wang et al., 2011; Ehn et al., 2014; Patoulias et al., 2015; Wang et al., 2015). Numerous lab experiments and quantum chemical calculations reported amines play an important role in atmospheric nucleation and the growth of new particles (Zhang et al., 2012; Jen et al., 2014; Chen et al., 2015a, b, 2017; Olenius et al., 2017), and oxalic acid, one of the most abundant dicarboxylic acids, was also proposed to be involved in NPF, and to be more readily bound to methylamine than to ammonia (Guo et al., 2016; Arquero et al., 2017; Hong et al., 2018). So far, it has been a challenge to accurately measure the mass concentrations of aminiums and oxalate in nucleation mode particles (Bzdek et al., 2013; Tao et al., 2016). During the sampling using the nano-MOUDI in this study, the bounce-off effect could be much large through a comparison between a rough estimation of mass concentration of nanometer particles in three size bins of 10-56 nm using the number concentration measured by FMPS and the sum of ionic concentrations therein (not shown). The derived errors on aminiums and oxalate in nanometer particles with aerodynamic diameter smaller than 56 nm was corrected by assuming that the observed $SO_4^{2-}$ therein was completely attributed to the error. The bounce-off error was thereby estimated by the concentration of $SO_4^{2-}$ measured in nanometer particles multiplying the highest ratios of aminium to $SO_4^{2-}$ and oxalate to $SO_4^{2-}$ in submicron size bins, i.e., either 0.32-0.56 μm or 0.56-1.0 μm, in the same day sample. The same comparison was conducted for nanometer particles between 56-100 nm (not shown) and the mass fraction of the sum of ionic species to the estimated total mass was reasonably consistent with those in submicron reported in literature. This suggested that the error appeared to be minor therein.

Based on the method mentioned above, the aminium and oxalic acid were corrected in 18-56 nm particles. The mass fractions of aminiums and oxalate in 18-100 nm particles on NPF days for Class-I and Class-II (discussed in Section 3.4) were shown in Fig. 9a-b. Here we exclude the analysis of 10-18 nm particles, due to the large underestimation of <10 nm (mobility diameter) particles number concentration measured by FMPS. In Class-I NPF events, appreciable contributions of oxalate and $TMA^+$ to total corresponding size particles mass were found in the size bin of 18-32 nm and 32-56 nm, with oxalate 5-10 times higher than $TMA^+$ in each of these two bins (Fig. 9a). For example, the mass concentration of oxalate in 18-32 nm particles was 10.0±10.0 ng m$^{-3}$, accounting for 13%±12% of the calculated particle mass concentration across the same size bin. In 32-56 nm and 56-100 nm particles, the mass concentration of oxalate was 10.9±8.2 ng m$^{-3}$ and 5.8±2.4 ng m$^{-3}$, accounting for 3%±3% and 1.6%±0.6% in each of the two size bins, respectively. The mass fraction of $TMA^+$ was 5%±8% in 18-32 nm particles (mass concentration of 2.4±1.9 ng m$^{-3}$), while it became minor in larger particles. In contrast, the $DMA^+$ was undetectable in 18-32 nm and 56-100 nm particles. There was appreciable $DMA^+$ in 32-56 nm particles, suggesting that $DMA^+$ might be involved in the growth of that size particles. In Class-II, the nanoparticles we collected were pre-existing particles or sourced from ocean emitted, rather than the newly formed particles. To more directly compare the mass concentrations of the two species between Class-I and Class-II, the absolute value was shown in Fig. 9c-d, noting that the day of 30 August 2015 was excluded considering the quite high concentration of $TMA^+$ sourced from oceanic precursors. The mass concentration of



TMA+ (black squares in shading area) and oxalate (red circles in shading area) in 10-56 nm particles in Class-I NPF events was significantly higher than that in Class-II NPF events, suggesting appreciable contribution of TMA and oxalic acid to the growth of new particles.

## 5 Conclusions

During seven cruise campaigns with 165 observational days, a total of 25 NPF events were observed in the atmospheres over the marginal seas of China and NWPO and their occurrence frequency showed distinct seasonal characteristics with the highest percentage of 28% in fall. The 24-hr air mass back trajectories implied that 24 out of the 25 NPF events might be associated with the long-range transport of continental pollutants, supported by the number concentrations of pre-existing particles larger than 30 nm exceeding 1-2 orders of magnitude of the clean marine background values. The 24-hr air mass back trajectory of one event, during which the net increase of new particles number concentration was 5-20 times smaller than others, suggested that the NPF appeared to be driven by oceanic precursors alone.

Over the NWPO, only two NPF events lasting from dozens of minutes to one hour were observed in the remote marine atmosphere, but the nucleation particle mode can be clearly detected therein during all 36 measurement days. Moreover, the percentage of the $N_{<30 \text{ nm}}$ out of the total number concentration was as high as 40±13% on non-NPF days, implying frequent NPF events might occur aloft in the atmosphere over the NWPO. However, clear NPF events can be observed in the sea-level atmosphere when certain criteria were satisfied, i.e., a strong downward transport of air masses occurs during the initial period of NPF events.

The moderately good linear correlations can be obtained between NMINP and FR, and between $D_{pgmax}$ and GR. The threshold of FRs was 8 cm$^{-3}$ s$^{-1}$, and the H$_2$SO$_4$ vapor was argued to be relatively sufficient compared with organic vapors in NPF events at FR<8 cm$^{-3}$ s$^{-1}$. At FR>8 cm$^{-3}$ s$^{-1}$, the H$_2$SO$_4$ vapor was argued to be the limiting factor for the net increase of new particles in number concentration and organic vapors were argued to determine FR.

We found that simultaneous measurements of particle number concentration and CCN at different values of SS can largely improve our understandings not only for the contribution of grown new particles to CCN, but also which chemicals determining the growth of new particles to CCN size. For example, during the NPF event on 4 September 2015, an increase of $N_{CCN}$ can be clearly observed when the $D_{pg}$ increased beyond 50 nm at the SS of 0.4%. However, no clear increase of $N_{CCN}$ can be observed during the whole event at the SS of 0.2%. This suggested that grown new particles can activate as CCN size at moderately high SS. The $D_{pgmax}$ was 77 nm and the SP was 0.83 during the NPF event, implying a large percentage of newly formed particles can grow to activate as CCN. However, in the remaining 24 NPF days, $D_{pgmax}$ was less than 50 nm and SP varied from 0 to 0.29, indicating that the probability of new particles being activated as CCN was low. Moreover, the combination analysis of CN and CCN during the NPF event on 4 September 2015 showed that the value of κ decreased from 0.4 to 0.1 with new particles growing to CCN size, implying that organics likely overwhelmed the growth.

The simultaneous measurements made in the atmosphere over the marginal seas and at OUC site showed regional NPF events occur in 12 days out of the total of 59 days. In the regional NPF events, the NPF event over the marginal seas was



featured by the larger FRs and comparable GRs. Three NPF events were observed over the marginal seas with no growth or extremely low growth, and the corresponding events at OUC showed no growth or extremely low growth in two days while new particles showed obvious growth in one day. No apparent growth or the extremely low growth of new particles in the marine atmospheres implied that semi-volatile species thermodynamically can not support the growth of >10 nm particles.

When new particles showed obvious growth in the marine atmosphere, oxalate and TMA$^+$ occupied the significant proportion in <56 nm particles, suggested that TMA and oxalic acid might yield an appreciable contribution to the growth of new particles.

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





**Figure 1** Cruise track over the marginal seas of China and NWPO. (a: Fall cruises during 16 October to 5 November 2011, 2-20 November in 2012 and 5-25 November in 2013, b: Spring cruise during 28 April - 19 May in 2014, c: Summer cruises during 18 August - 5 September in 2015 and 28 June - 20 July in 2016, d: NWPO cruise during 18 March - 22 April in 2014. Pentacles represent the ship location of occurring NPF events).



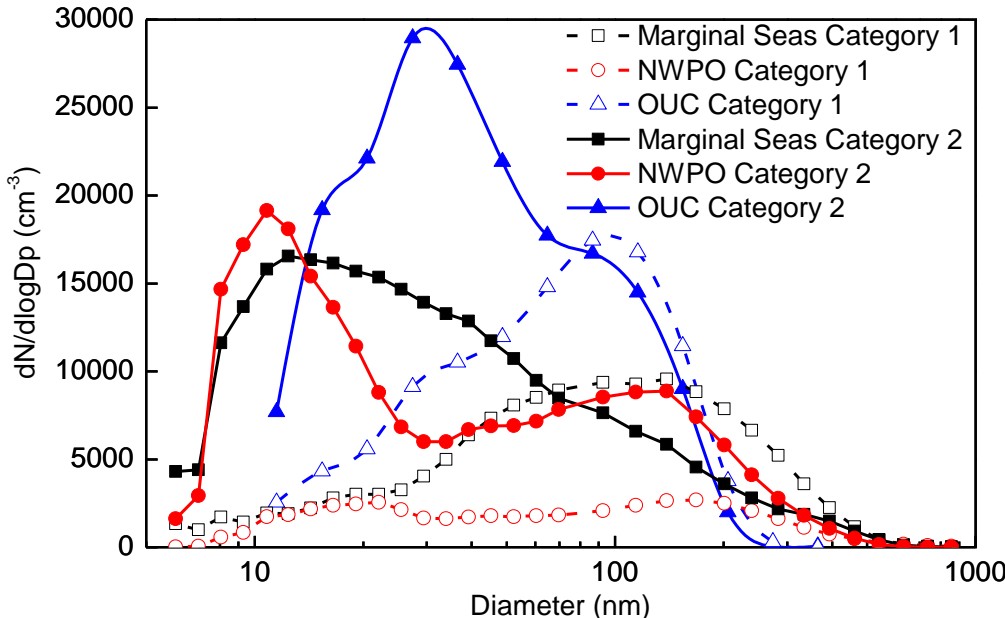

**Figure 2** Averaged particle number size distributions in the atmospheres over the marginal seas of China, NWPO and at OUC site (Category 1 represents background particle signals and Category 2 represents new particle signals).







**Figure 3** Relationship between new particle formation rate (FR) and net maximum increase in nucleation mode particles number concentration (NMINP), growth rate (GR) and maximum geometric median diameter of new particles ($D_{pgmax}$) in NPF events occurred over the marginal seas of China and NWPO (Black symbols were treated as exteriors and excluded from the correlation analysis).





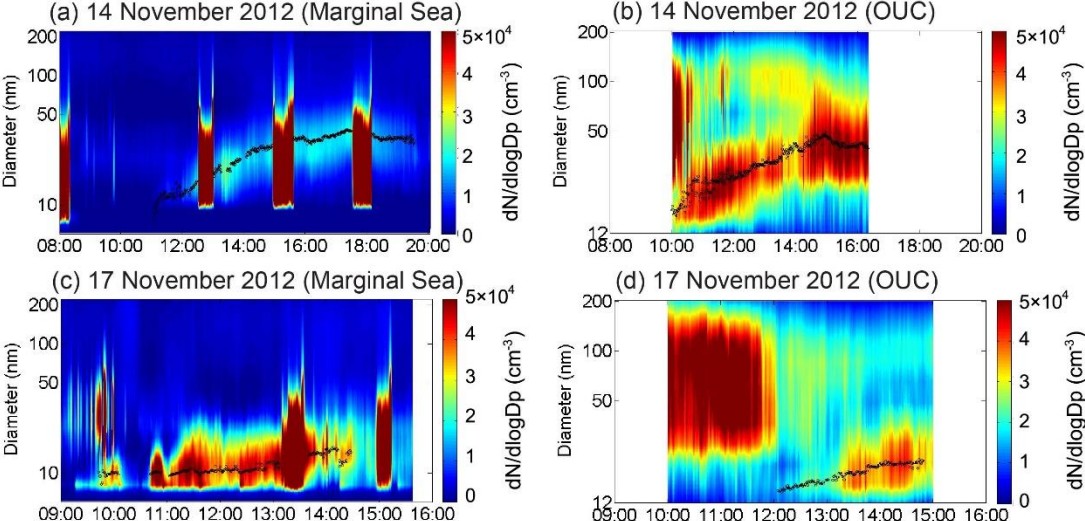

**Figure 4** Class-I (a, b) and Class-II (c, d) NPF events in the atmosphere over marginal seas of China and at OUC site.





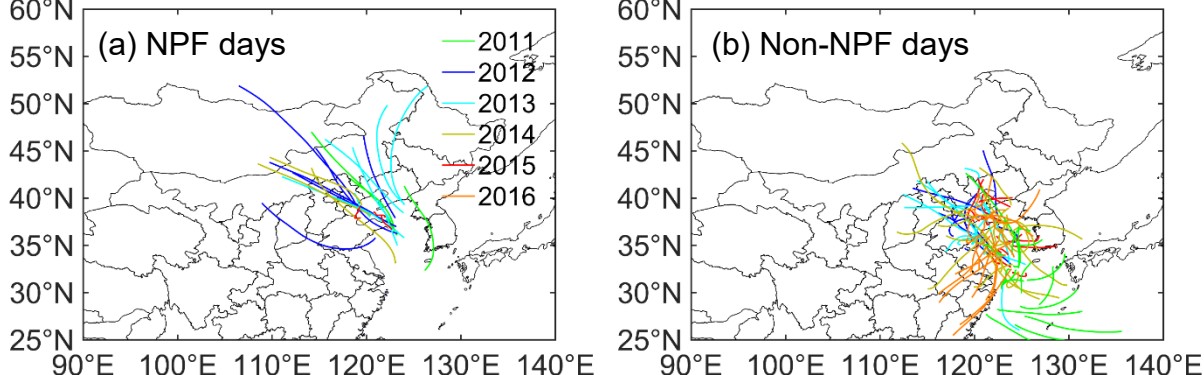

**Figure 5** 24-hr air mass back trajectories in the atmosphere over marginal seas of China (the simulation time was the NPF initial time on NPF days and 8:00 A.M. (UTC+8) on non-NPF days).



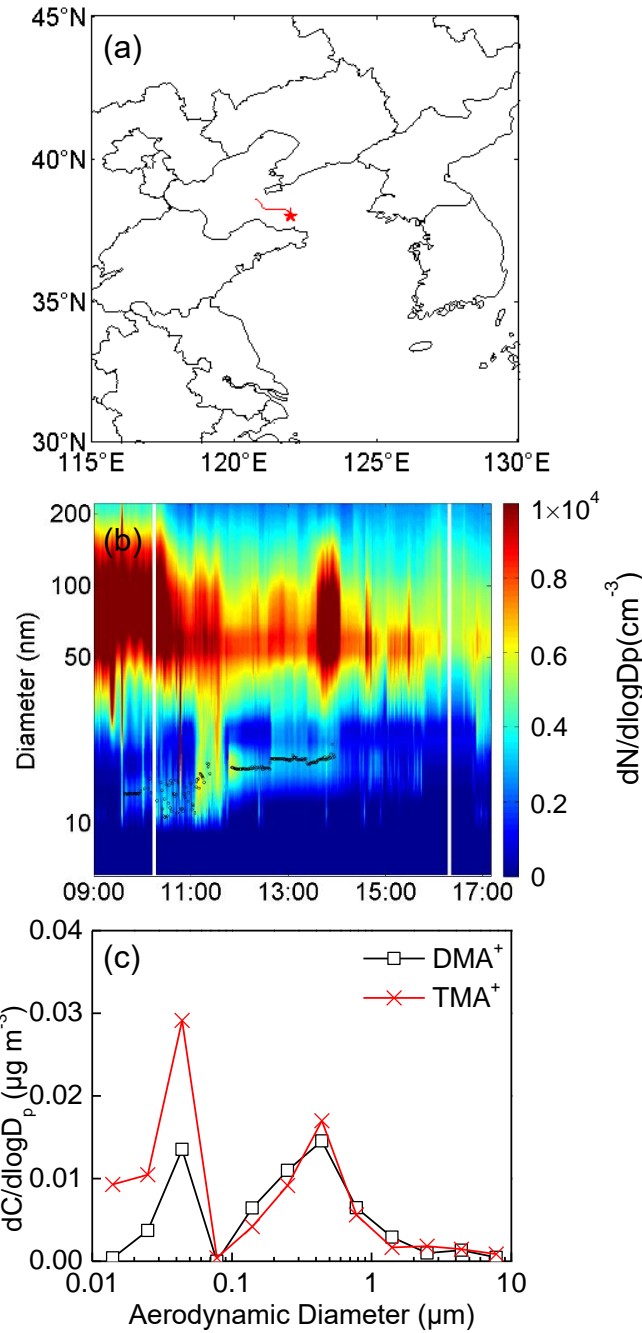

**Figure 6** 24-h air mass back trajectory, contour plot of NPF events and size distributions of particulate DMA[+], TMA[+] (sampling period was 7:44 - 21:15) on 30 August 2015.





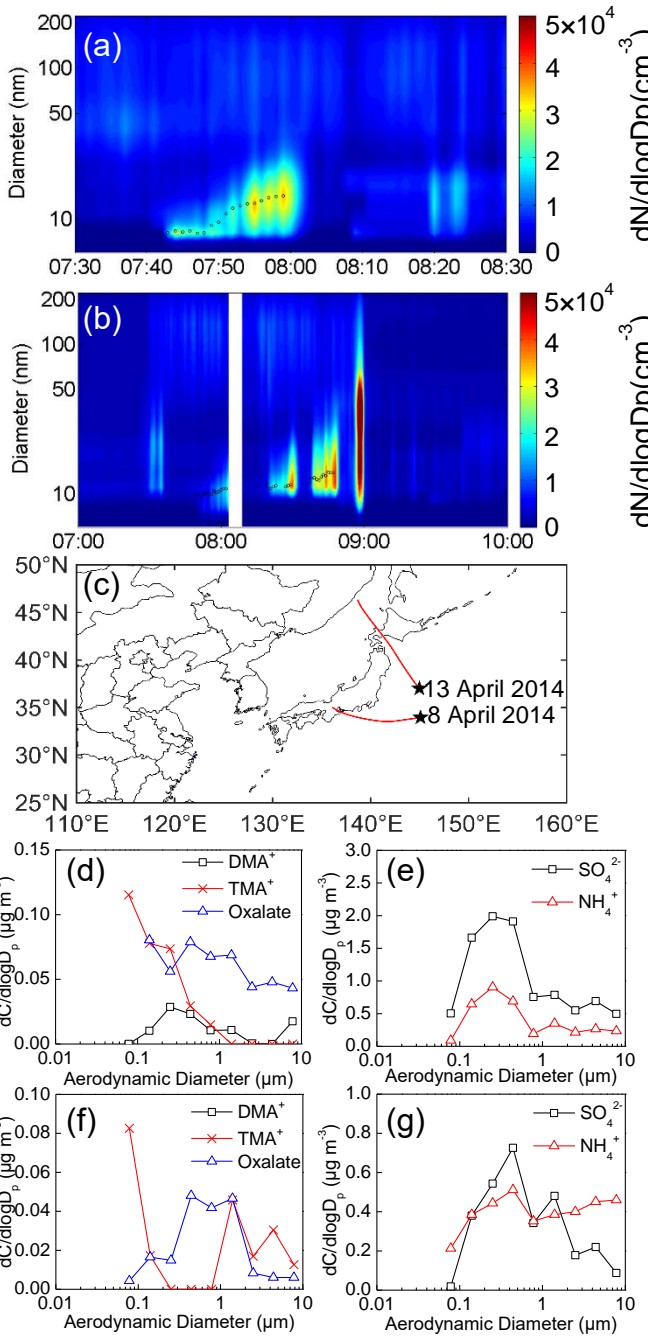

**Figure 7** NPF events on 8 April 2014 (a) and 13 April 2014 (b), 24-h air mass back trajectories (c), and size distributions of particulate DMA$^+$, TMA$^+$, oxalate, SO$_4^{2-}$, NH$_4^+$ in mass concentration on 8 April 2014 (d, e) and 13 April 2014 (f, g) (sampling periods were 11:12-23:33 on 8 April 2014 and 9:10- 21:05 on 13 April 2014).



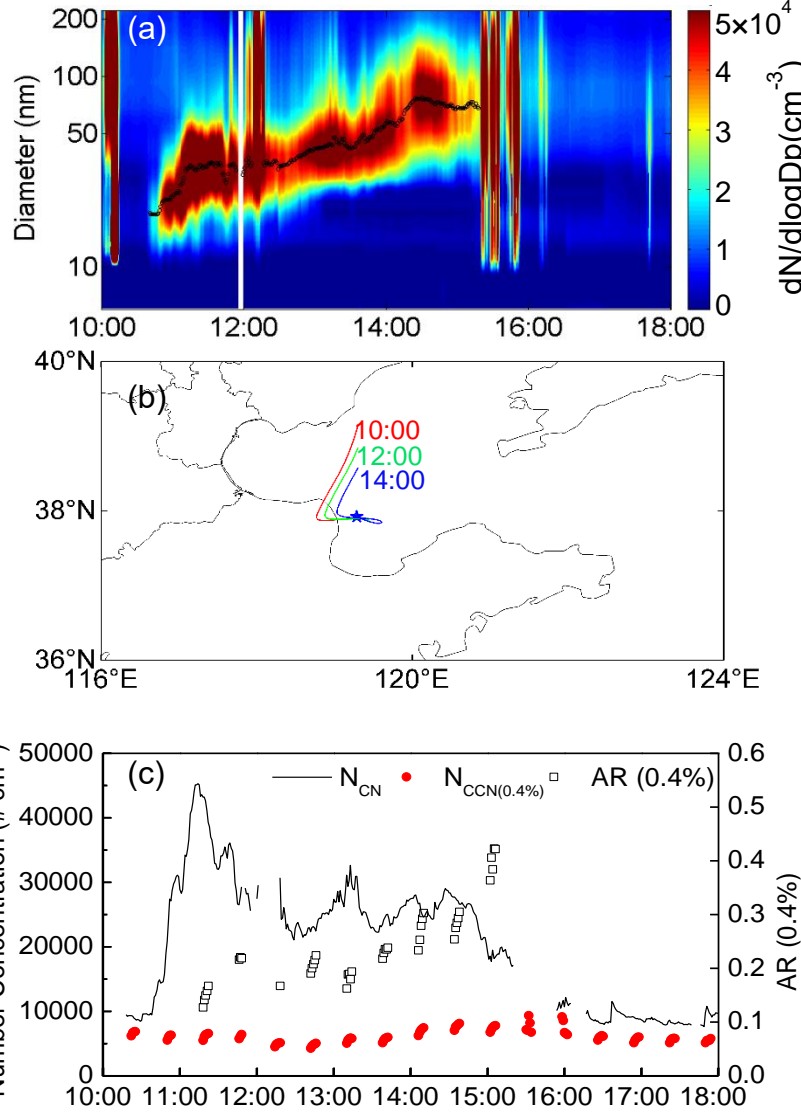

**Figure 8** Contour plot of NPF event, 24-h air mass back trajectory throughout the NPF event, and the time series of total particle number concentration ($N_{CN}$), CCN number concentration at the SS of 0.4% ($N_{CCN(0.4)}$), and CCN activation ratio (AR) on 4 September 2015 in BS.



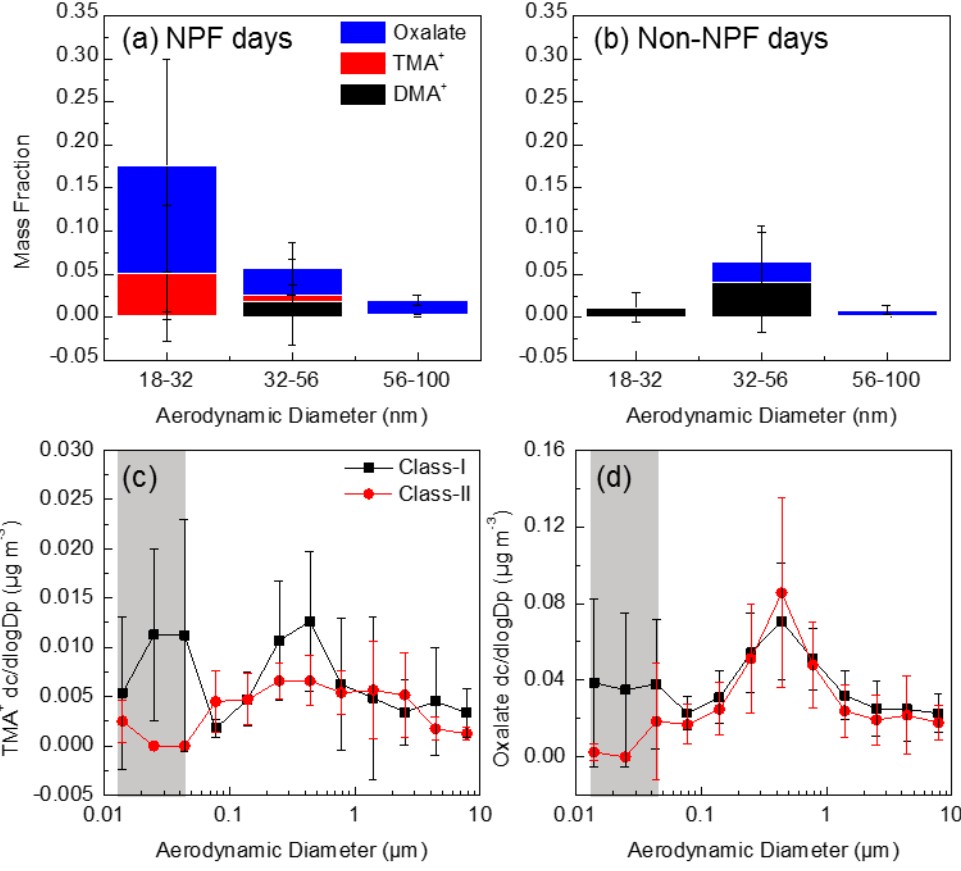

**Figure 9** Averaged mass fraction of aminiums and oxalate in 18-30 nm, 32-56 nm and 56-100 nm particles on NPF days and non-NPF days (a, b) and the averaged size distributions of TMA$^+$ and oxalate on Class-I and Class-II NPF days.



**Table 1 Summary of NPF frequency, FR, GR, CS, NMINP, and $D_{pgmax}$ over the marginal seas of China, NWPO and at OUC site**

| Campaigns | Location | Frequency | FR ($cm^{-3}s^{-1}$) | GR ($nm\ h^{-1}$) | CS ($10^{-2}\ s^{-1}$) | NMINP ($10^{4}\ cm^{-3}$) | $D_{pgmax}$ (nm) | Regional NPF event |
|---|---|---|---|---|---|---|---|---|
| Fall Cruise Oct. 16-Nov. 15 2011 Nov. 2-20 2012 Nov. 5-25 2013 | Marginal Seas | 28% (17/61days) | 5.3±5.4 | 3.5±2.1 | 0.8±0.3 | 1.6±1.0 | 50 | 11 days |
| | OUC | 30% (12/40days) | 2.4±1.1 | 4.3±2.2 | 4.1±2.1 | 0.9±0.4 | 100 | |
| Summer Cruise Aug.18-Sep.5 2015 Jun.28-Jul.20 2016 | Marginal Seas | 7% (3/43days) | 10.5±9.3 | 7.3±4.9 | 2.7±0.7 | 1.4±1.1 | 77 | 1 day |
| | OUC | 22% (4/18days) | 3.2±2.3 | 11.3±6.8 | 1.2±0.3 | 0.5±0.3 | 45 | |
| Spring Cruise Apr.28-May.19 2014 | Marginal Seas | 14% (3/22days) | 6.9±3.8 | 4.2±2.2 | 2.0±0.5 | 1.8±1.2 | 50 | |
| NWPO Cruise Mar.18-Apr.22 2014 | NWPO | 6% (2/36days) | 15.7±7.9 | 15.0±16.1 | 1.3±0.9 | 1.3±0.5 | 14 | |
| Average All | Marginal Seas + NWPO | 15% (25/162days) | 6.4±5.8 | 4.7±4.7 | 1.3±0.9 | 1.6±0.9 | 77 | 12 days |
| | OUC | 29% (17/59days) | 2.7±1.5 | 6.1±4.8 | 3.2±2.2 | 0.8±0.4 | 100 | |

[a]: Condensation sink (CS) was averaged 1-h prior to the NPF events.



# Appendices

**Appendix A** General information of NPF events.

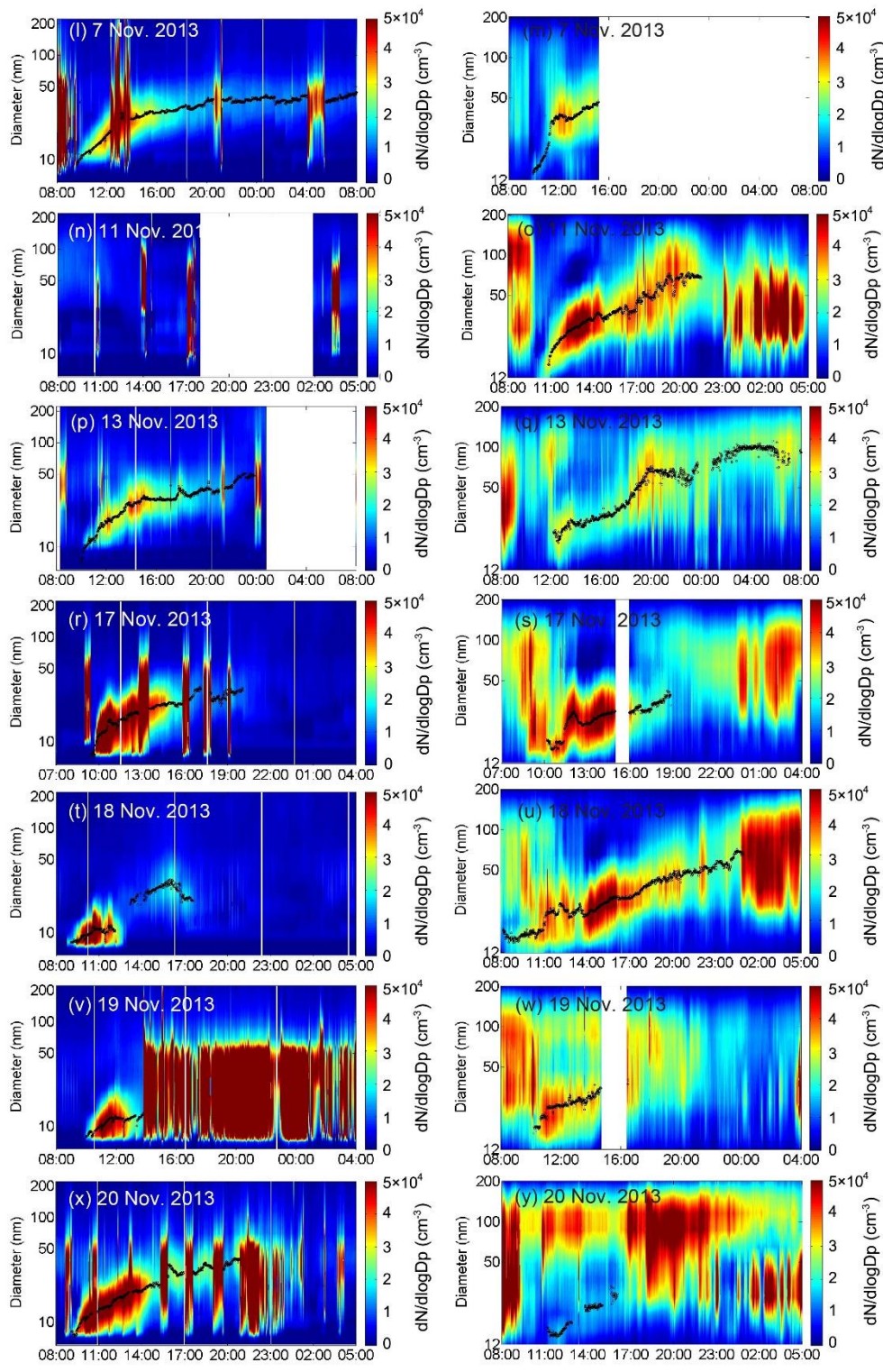











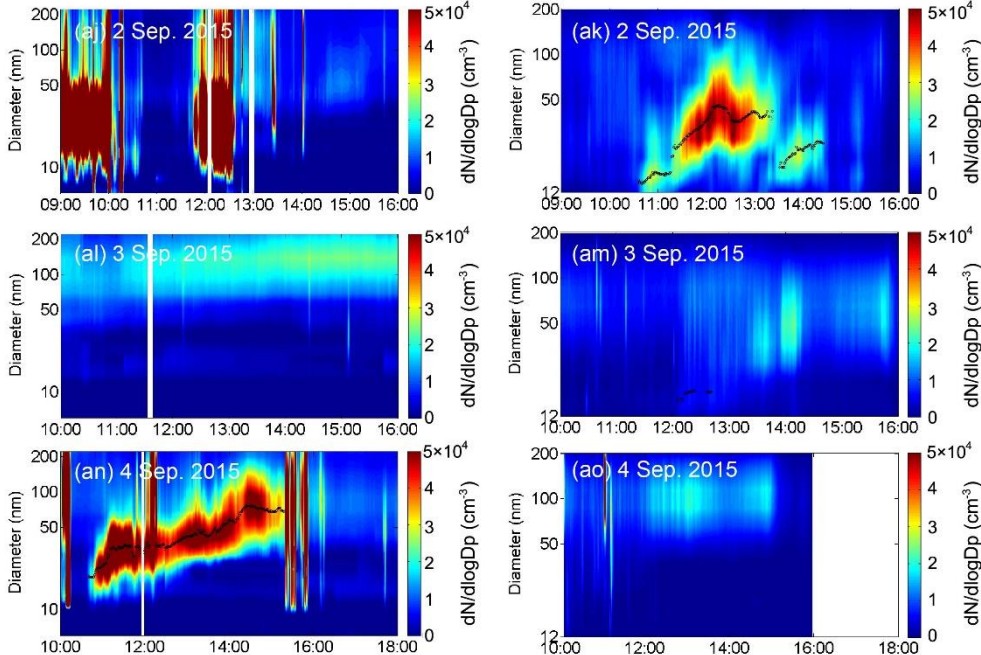

**Figure A1** Contour plot of NPF events in the atmospheres over marginal seas of China and at OUC site (Left panels represent over the marginal seas, right panels represent the simultaneous measurement at OUC site. The contour plot of NPF events in 2011 can be found in Liu et al. (2014), and no NPF events were observed during the cruise in 2016).



**Table A1** Characteristics of NPF events in atmospheres over marginal seas of China (SYS, NYS, ECS, BS), NWPO and at OUC site.

| Date | Classification | Location | FR (cm$^{-3}$s$^{-1}$) | GR (nm h$^{-1}$) | CS (10$^{-2}$ s$^{-1}$)[a] | NMINP (cm$^{-3}$) | D$_{pgmax}$ (nm) | SP |
|---|---|---|---|---|---|---|---|---|
| 17 October 2011* | Class-I | ST1 in SYS | 15.2/4.1 | 2.5/7.5 | 0.4±0.0 | 17490/ 14614 | 42/50 | 0.03 |
| 18 October 2011* | Class-I | ST2 in SYS | 7.5 | 3.5 | - | 35909 | 28 | 0 |
| 19 October 2011* | Class-I | ST3 in SYS | 0.3/1.1 | 3.4 | 0.7±0.0 | 438/ 3143 | 22 | 0 |
| 26 October 2011* | Class-I | ST4 in ECS | 1.6 | 4.4 | 0.6±0.0 | 6691 | 21 | 0 |
| 4 November 2012* | Class-I | ST5 in SYS | 1.4/3.1 | 5.0/10.0 | - | 8627/ 14083 | 39/47 | 0.10 |
|  |  | OUC | - | 5.5 | - | - | 40 | 0.14 |
| 12 November 2012 | Class-II | ST6 in SYS | - | 0.2 | - | - | 12 | 0 |
|  |  | OUC | 3.5 | 1 | 5.3±0.6 | 4676 | 17 | 0 |
| 13 November 2012 | Class-I | ST7 in SYS | 7.2 | 3.0/3.0 | 1.1±0.0 | 19916 | 28/34 | 0 |
| 14 November 2012 | Class-I | ST8 in NYS | 1.1(0.71) | 4.4 | 0.3±0.1 | 5588 | 38 | 0.2 |
|  |  | OUC | 2.8 | 5.7 | 6.5±1.1 | 7089 | 49 | 0.27 |
| 17 November 2012 | Class-II | ST9 in BS | 3.2/18.8 (2.3/16.4) | 1.0 | 1.3±0.1 | 14420/ 19717 | 15 | 0 |
|  |  | OUC | 2.7 | 3.1 | 7.5±1.3 | 10518 | 20 | 0 |
| 19 November 2012 | Class-I | ST10 in BS | - | 4.5/1.7 | - | - | 30/50 | 0.16 |
|  |  | OUC | 4.8 | 8.9 | - | 14248 | 49 | 0.6 |
| 7 November 2013 | Class-I | ST11 in SYS | 1.9(1.8) | 2.4 | - | 10458 | 42 | 0.09 |
|  |  | OUC | 0.9 | 6 | 1.7±0.2 | 6571 | 46 | 0.28 |
| 11 November 2013 | Class-I | OUC | 1.8 | 5.4/6.0 | 1.2±0.3 | 12048 | 37/72 | 1.12 |
| 13 November 2013 | Class-I | ST12 in SYS | 2.1(2.1) | 4.1/3.3 | 1.1±0.1 | 17005 | 30/49 | 0.11 |
|  |  | OUC | 2 | 2.3/6.9 | 2.6±0.6 | 10452 | 30/100 | 1.76 |





| 17 November 2013 | Class-I | ST13 in NYS | 15.6(9.9) | 2.5/3.5 | 0.5±0.0 | 31391 | 32/32 | 0 |
|---|---|---|---|---|---|---|---|---|
| | | OUC | 2.6 | 2 | 4.2±0.8 | 5458 | 38 | 0.06 |
| 18 November 2013 | Class-I | ST14 in NYS | 3.4(1.7) | 3.5 | 0.7±0.1 | 22613 | 30 | 0.03 |
| | | OUC | 2.2 | 2.0/3.9 | 3.0±0.3 | 12719 | 30/70 | 0.56 |
| 19 November 2013 | Class-I/II | ST15 in NYS (II) | 3.8(2.3) | 1.1 | 0.6±0.1 | 23886 | 12 | 0 |
| | | OUC (I) | 2.3 | 3 | 5.3±0.2 | 10556 | 40 | 0 |
| 20 November 2013 | Class-I | ST16 in NYS | 6.6(4.3) | 2.8 | 0.8±0.1 | 28818 | 40 | 0.03 |
| | | OUC | 1.3 | 3.5 | 3.4±0.1 | 3032 | 28 | 0 |
| 21 November 2013 | Class-II | ST17 in BS | 3.2 | 3.4 | 1.2±0.2 | 24845 | 17 | 0 |
| 5 May 2014 | Class-II | ST18 in SYS | 2.5 | 3.3 | 1.7±0.2 | 4163 | 14 | 0 |
| 14 May 2014 | Class-I | ST19 in BS | 9.3 | 6.7 | 1.7±0.4 | 31545 | 50 | 0.29 |
| 15 May 2014 | Class-II | ST20 in BS | 8.8 | 2.5 | 2.5±0.2 | 25746 | 13 | 0 |
| 27 August 2015 | Class-I | ST21 in SYS | 18.5(13.3) | 9.6 | 3.5±0.5 | 18930 | 41 | 0.05 |
| | | OUC | 3.2 | 16.2 | 1.4±0.1 | 6720 | 25 | 0 |
| 28 August 2015 | Class-I | OUC | 2.1 | 8.0 | 1.3±0.2 | 4873 | 26 | 0 |
| 30 August 2015 | Class-II | ST22 in NYS | 0.3 | 1.7 | 2.1±0.2 | 1450 | 19 | 0 |
| 2 September 2015 | Class-I | OUC | 6.7/3.5 | 20.3/9 | 1.2±0.2 | 9034/ 2915 | 45/26 | 0.91 |
| 3 September 2015 | Class-II | OUC | 0.5 | 3.2 | 0.8±0.1 | 1847 | 18 | 0 |
| 4 September 2015 | Class-I | ST23 in BS | 12.7 | 10.6 | 2.5±0.0 | 22355 | 77 | 0.83 |
| 8 April 2014 | Class-II | ST24 in NWPO | 11.8 | 26.3 | 1.9±0.2 | 11709 | 14 | 0 |
| 13 April 2014 | Class-II | ST25 in NWPO | - | 3.6 | 0.6±0.1 | - | 14 | 0 |





ᵃ: Condensation sink (CS) was averaged 1-h prior to the nucleation event.

*: Liu et al., 2014.

(): particles with diameter larger than 10 nm were used to calculate FRs for FMPS data when the parallel measurements were conduct.