# Peer review of "New particle formation in marine atmosphere during seven cruise campaigns"

_Atmospheric Chemistry and Physics, 2018_

## Referee Comment (RC1) · Anonymous Referee #1 · 15 Aug 2018

This manuscript combines measurements conducted over several marine cruises to investigate atmospheric new particle formation (NPF) and growth in the marine atmosphere. The paper appears scientifically sound and original enough to merit publications. In its current form, the paper requires, however, important revisions, especially what it comes to the technical quality of the paper.

Scientific issues

The second paragraph of Introduction gives a background on NPF in the marine atmosphere. It contains a sentences discussing the role of amines in NPF (lines 22-23 on page 2) which is no way related to marine NPF. I recommend this sentence to be removed from here. The discussion on role of ions in coastal NPF does not include the paper by Sipila et al (2016, Nature) that gives the most detailed molecular view on this

process published so far.

Page 6, line 9: the numbers appear too accurate. I suggest writing: . . .event to be at least 50-500 km.

Page 6, line 26: the selected border between the Aitken and accumulation mode (50 nm) is very untypical. Normally in a scientific literature, it is assumed to be between 80 and 100 nm. Please correct or give a reason for this choice.

Page 11, lines 7-8: this statement requires a couple of more, and more recent, references.

Page 13, lines 12-13: I am confused about this assumption. Do you mean that there should be no sulfate in nm sized particles?

Page 14, line 12: dozens of minutes to one hours sounds a very strange range because dozens corresponds to several tens of minutes and one hour is the same (60 min). Please correct or modify.

The paper has several sentences that are either difficult to understand or written in bad style, so they need to be rewritten. They are in the following places: page 4, lines 1-2; page 5, lines 30-31; page 7, lines 20-22; page 7, lines 27-31; page 8, lines 23-29; page 9, lines 2-3; page 9, lines 24-26; page 9, lines 29-30; page 10, lines 1-5; page 11, line 4-5; page 11, lines 12-15; page 12, lines 11-13; page 14, lines 10-11; page 10, line 14-15; page 14, lines 22-28.

Technical issues

The paper refers to figures and tables marked as S1, S2 etc. They are in Appendix, so A1, A2 etc would be more logical way to refer to them.

The following grammatical corrections are needed (the text below give the correct way to write them):

page 3, line 32: . . .monsuun prevails. . .

page 6, line 10: ...sinks are two...

page 6, line 20: ...first classified...

page 6, line 26: We first discuss category I data over the marginal...

page 7, line 1: ...lower than that over the marginal...

page 7, line 2 ...as over the marginal seas (20%), indicating ...

page 7, line 5: ...altitudes

page 7, line 9: ...with diameters lower than 20 nm

page 7, line 11: ...higher than that reported in previous...

page 7, line 16: a comparable

page 7, line 27: over the...during the three...

page 8, line 1: ...with a high..

page 8, lines 2-4: ...intermittent occurrence of nucleation...here were much higher than those observed in previous... Altogether, considering both...

page 8, line 6: we next compare...

page 8, line 8: ...larger mean values

page 8, line 9: ...whereas comparable...

page 8, line 10: over the marginal...

page 8, lines 12-13: precursors, such as...vapors, were...

page 8, line 17: ...no obvious

page 9, line 1: ...24 NPF days, except on one day when it was 77 nm. ...could be identified

page 9, line 5: ...were able to grow...

page 9, line 18: ...event occur in regional

page 9, lines 19-20: ...event are mostly local phenomena reported in a few studies made over...

page 9, line 21: ...in an urban...

page 10, line 11: ...were accompanied

page 10, line 16: ...zoomed in (Fig. 6a).

page 10, line 17: ..high relative humidity of 74% and low wind speed of ..

page 10, line 18: ...characterized by a low

page 10, line 19: during the first hour

page 10, line 21-22: ...during the first 30 minutes...fluctuated...during the following 3 hours...

page 10, line 28: ...suggests a strong

page 11, line 3: and it lasted...the total particle

page 11, line 16: ..compounds may be involved in

page 11, line 17: day was analyzed

page 11, line 18: particles smaller than 10

page 11, line 19: ...(derived...respectively, higher than in other

page 11, line 22: ...involved in... moderately high

page 11, line 31: lower than

page 12, line 6: ...at an initially high relative humidity of

page 12, line 17: we found that the

page 12, line 25-26: implying that the majority of . . . particle were able to grow to CCN at

page 12, line 28: to act as CCN

page 13, line 5: to play

page 13, line 12: errors in

page 14, line 18: Moderately good. . . were obtained

Finally, please check out carefully the language of the abstract.

---

## Referee Comment (RC2) · Anonymous Referee #2 · 16 Aug 2018

Overall comment: This study includes the observations of nanoparticles in six cruises over the marginal seas of China and one cruise to the Northwest Pacific Ocean. The particle number concentration, size distribution, formation rate and growth rate of new particles are discussed. The authors also try to illustrate the roles of anthropogenic and marine biogenic emissions in new particle formation, through analyses on several specific NPF events. The experiments are interesting, and should be beneficial to advance the knowledge on the impacts of human being activities on NPF and global climate change. However, the experimental design has obvious drawbacks in considering the adequate data to support the analyses in this meaningful research. Nearly no data of the precursors of condensable vapors are available. Though some chemicals, such as the amines and the oxalic acid, in the size-segregated are analyzed, the

sampling period even missed the NPF periods, which led to the inappropriateness of using these data to infer the processes and chemical species dominating NPF. I also have serious concern on the explanations to the different relationships between the formation rate and the net maximum increase in the nucleation mode particle number concentration. Similarly, the conclusion that the NPFs, regardless of which categories, are regional phenomenon cannot convince me, since no solid evidence has been provided. In view of the inadequate discussions, misleading inferences and even wrong interpretations, the paper needs to be revised substantially before being considered to be accepted. Specific comments are also given for the authors' reference. Specific comments: 1. Page 3, "In November 2012, the NO2 column densities were higher in the eastern mainland of China due to the house-heating". House heating is not the sole cause of elevated NO2 in autumn. 2. Page 6, lines 6-10. How do you confirm that these NPF events were the regional NPF events, rather than the local ones that occasionally occurred on the same days? Is there any evidence proving that the air masses were homogeneous on these days, except for the backward trajectories? Since the ship location and the coastal sites were generally in an area influenced by the same monsoon, they always received air masses from the same directions. However, it does not mean that the regional air overrode the properties of local air masses. 3. Page 6, lines 11-13. From the particle number distributions shown in Fig. A1h, i, l, m, I can hardly believe that these are the regional NPF events. Besides, could the delay be caused by the different weather conditions, or downward transport of nanoparticles in the afternoon? 4. Section 3.2. The observational particle number distributions at OUC were not well presented. 5. Page 6, lines 20-23. How did you remove the influence of ship-self emissions? This needs to be demonstrated in methodology. 6. Page 7, lines 13-14. "The increase likely induced by the long-range transport of air pollutants from the continents, inferred from the doubled number concentrations of accumulation mode particles in Category 2 relative to Category 1." This is contradictory to the previous statement that "the concentration increase was limited to particles with the diameter less than 20 nm". 7. Page 7, lines 19-22. The authors should illustrate in more

details the size ceilings that the particles could grow up to. What caused the different ceilings, and what were the implications from the differences in particle size distributions? 8. Page 8, lines 10-15. Condensation sink is an important factor influencing particle formation. Throughout the manuscript, CS has never been presented and has seldom referenced for discussions. The lack of measurements of condensable vapors makes so many inferences in the paper not reliable, not to say some inferences are contradictory to common sense. For example, here I cannot believe that the loadings of precursors favorable for the formation of new particles were higher over the marginal seas than in the coastal area. Evidences need to be provided to support the inferences. 9. Page 8, lines 17-32. I cannot understand why the higher formation rate did not result in larger increase of nucleation mode particles, note that the formation rate is closely related to the increase of nucleation mode particles if looking at the calculation formula of formation rate. All the explanations are based on the assumptions, which cannot convince me. The authors should provide more evidences to validate their assumptions. The authors state that "the NMINP was always determined by the consumed H2SO4 vapor for nucleation". Sorry for that I cannot accept this view. How about the number of nucleation mode particles when the organic vapors facilitated the nucleation and particle growth to the detectable size? The so called threshold of formation rate, i.e. 8 cm-3s-1, was exactly the same as that reported in the study previously published by the same authors. This cannot convince the readers unless the similar phenomenon has been reported by other groups. I tried to understand the authors' view by finding the clues from the paper "Simultaneous measurements of new particle formation at 1 s time resolution at a street site and a rooftop site". However, it is hard for me to follow up the authors in many points. For example, in this paper, the sentence "Supposing that sulfuric acid vapors are completely nucleated, followed by the nucleated particles growing to the detectable size, the yields of newly formed particles are determined mainly by the supply of sulfuric acid vapor and are less affected by the formation rate" is problematic. How could you separate the role of sulfuric acid from the formation rate, as sulfuric acid plays critical role in nucleation? In the sentence "Scenario 1: H2SO4

vapor is relatively sufficient against NucOrg, and J8 is therefore determined mainly by the availability of NucOrg vapor. A good correlation is theoretically expected for J8 and NMINP". To be honest, I do not understand the logics behind. 10. Page 9, lines 29-33. The concurrent occurrences of class II NPF events at the coastal site and over the marginal seas could not be an evidence of the regional characteristics. The particle number distributions at the two sites were quite different on the days specified by the authors (Figure A1). Besides, it is difficult to convince me with the backward trajectories. The two sites were in a same region under the influence of the same monsoon. Even so, the air masses could be totally different in chemical compositions when they passed over the different cities. With no chemical information or mesoscale simulation, it is hard to say the two sites were interacted and the regional NPF events occurred at the two sites. 11. Page 10, lines 1-6. Condensable vapors are of course critical in NPF. However, it is not reasonable to simply attribute the different characteristics of NPF to the abundances of the condensable vapors. Other factors, such as the preexisting particles and the meteorological conditions also influence the NPF. In this case, more preexisting particles with larger diameters existed at the marginal sea site. Could this also account for the insignificant particle growth? 12. Section 4.1. I do not agree that new particle formation occurred in this case, i.e. 30 August 2015. 13. Page 11, lines 12-15. Figure 7c does not show the altitude variation of the backward trajectories. 14. Page 11, lines 16-26. The sampling periods of MOUDI samples were after the NPF events, not including the hours when the new particles were formed and grew up. I would doubt the reasonability of using these data to infer the chemical species dominating NPF. Same for the other similar discussions. 15. Page 12, lines 8-10. I do not understand the logics behind this inference, though it is true that the AR increased after Dpg was higher than 50 nm. Why not present the number concentration of >50 nm particles or its fraction in total particles against the NCCN? It would be a more direct way to link the particles larger than 50 nm to CCN. 16. Caption of Figure 3, what does "exteriors" mean? Why should they be excluded from the regression? Figure 4, what does the black dots represent, same for the other figures? Figure 9, what does the

highlighted area denote for? 17. The manuscript needs to be grammatically checked by an editing company or a native English speaker professor.

---

## Author Comment (AC1) · 5 Nov 2018

This manuscript combines measurements conducted over several marine cruises to investigate atmospheric new particle formation (NPF) and growth in the marine atmosphere. The paper appears scientifically sound and original enough to merit publications. In its current form, the paper requires, however, important revisions, especially what it comes to the technical quality of the paper.

Response: The authors thank the reviewer's comments and try our best to respond and revise our manuscript accordingly.

Scientific issues

[Figure]

The second paragraph of Introduction gives a background on NPF in the marine atmosphere. It contains a sentences discussing the role of amines in NPF (lines 22-23 on page 2) which is no way related to marine NPF. I recommend this sentence to be removed from here. The discussion on role of ions in coastal NPF does not include the paper by Sipila et al (2016, Nature) that gives the most detailed molecular view on this process published so far.

Response: Ocean is one of the important sources of atmospheric amines. The authors will revise the sentence to "Moreover, amines, which can be produced through excretion and metabolism by a variety of marine organisms, were reported to enhance H2SO4-H2O nucleation and promote the growth of newly formed particles". The reference of Sipilä et al. (2016) will be added in the revised manuscript.

Page 6, line 9: the numbers appear too accurate. I suggest writing: . . .event to be at least 50-500 km.

Response: We corrected the sentence accordingly.

Page 6, line 26: the selected border between the Aitken and accumulation mode (50 nm) is very untypical. Normally in a scientific literature, it is assumed to be between 80 and 100 nm. Please correct or give a reason for this choice.

Response: Agree. On basis of those highly cited references, e.g., Kittelson, 1998, Kulmala et al., 2004, Kumar, et al., 2010; Seinfeld and Pandis, 2012, we revised the sentence as "The Aitken mode (30-100 nm) and accumulation mode (100-500 nm) were usually overlapped at the size range of 30-500 nm, with a minor nucleation mode at sizes below 30 nm".

Page 11, lines 7-8: this statement requires a couple of more, and more recent, references.

Response: In revision, the authors added the reference of Buzorius et al. (2004, J. Geophys. Res.), Quinn and Bates (2011, Nature) and Meng et al. (2015, Atmos.

[Figure]

Environ.), which reported that new particles can be formed above the marine boundary layer and mixing downward.

Page 13, lines 12-13: I am confused about this assumption. Do you mean that there should be no sulfate in nm sized particles?

Response: In the revision, we added "Note that appreciable amount of SO42- should exist in <56 nm particles, but the amount might be probably much smaller than sampling artifacts."

Page 14, line 12: dozens of minutes to one hours sounds a very strange range because dozens corresponds to several tens of minutes and one hour is the same (60 min). Please correct or modify.

Response: The authors revised as "from 17 minutes to one hour" in revision.

The paper has several sentences that are either difficult to understand or written in bad style, so they need to be rewritten. They are in the following places: page 4, lines 1-2; page 5, lines 30-31; page 7, lines 20-22; page 7, lines 27-31; page 8, lines 23-29; page 9, lines 2-3; page 9, lines 24-26; page 9, lines 29-30; page 10, lines 1-5; page 11, line 4-5; page 11, lines 12-15; page 12, lines 11-13; page 14, lines 10-11; page 10, line 14-15; page 14, lines 22-28.

Response: The authors have rewritten these sentences and double-checked the language using a profession service.

Technical issues

The paper refers to figures and tables marked as S1, S2 etc. They are in Appendix, so A1, A2 etc would be more logical way to refer to them.

Response: The original paper contain the Appendix in text and the Supplementary as attachment. Figure and table in appendix marked as Fig. A1 and Table A1, and the figures and tables in supplementary marked as Fig. S1 and table S1. We have double

check it.

The following grammatical corrections are needed (the text below give the correct way to write them):

page 3, line 32:. . .monsuun prevails. . .

page 6, line 10:. . .sinks are two. . .

page 6, line 20: . . .first classified. . .

page 6, line 26: We first discuss category I data over the marginal. . .

page 7, line 1: . . .lower than that over the marginal. . .

page 7, line 2 :. . .as over the marginal seas (20%), indicating. . .

page 7, line 5: . . .altitudes

page 7, line 9:. . .with diameters lower than 20 nm

page 7, line 11: . . .higher than that reported in previous. . .

page 7, line 16: a comparable

page 7, line 27: over the. . .during the three. . .

page 8, line 1:. . .with a high. . .

page 8, lines 2-4: . . .intermittent occurrence of nucleation. . .here were much higher than those observed in previous. . .Altogether, considering both. . .

page 8, line 6: we next compare. . .

page 8, line 8:. . .larger mean values

page 8, line 9:. . .whereas comparable. . .

page 8, line 10: over the marginal. . .
page 8, lines 12-13: precursors, such as...vapors, were...

page 8, line 17: ...no obvious

page 9, line 1: ...24 NPF days, except on one day when it was 77 nm. ...could be identified

page 9, line 5: ...were able to grow...

page 9, line 18:...event occur in regional

page 9, lines 19-20:...event are mostly local phenomena reported in a few studies made over...

page 9, line 21:...in an urban...

page 10, line 11: ...were accompanied

page 10, line 16: ...zoomed in (Fig. 6a).

page 10, line 17:...high relative humidity of 74% and low wind speed of ...

page 10, line 18: ...characterized by a low

page 10, line 19: during the first hour

page 10, line 21-22: ...during the first 30 minutes...fluctuated...during the following 3 hours...

page 10, line 28: ...suggests a strong

page 11, line 3: and it lasted...the total particle

page 11, line 16: ...compounds may be involved in

page 11, line 17: day was analyzed

page 11, line 18: particles smaller than 10

page 11, line 19:. . .(derived. . .respectively, higher than in other

page 11, line 22: . . .involved in. . . moderately high

page 11, line 31: lower than

page 12, line 6: . . .at an initially high relative humidity of

page 12, line 17: we found that the

page 12, line 25-26: implying that the majority of. . . particle were able to grow to CCN at

page 12, line 28: to act as CCN

page 13, line 5: to play

page 13, line 12: errors in

page 14, line 18: Moderately good. . .were obtained

Finally, please check out carefully the language of the abstract

Response: The authors thank the reviewer's grammatical comments. We corrected all of the errors above and double-checked the language using a profession service throughout the manuscript.

References

Buzorius, G., McNaughton, C. S., Clarke, A. D., Covert, D. S., Blomquist, B., Nielsen, K., and Brechtel, F. J.: Secondary aerosol formation in continental outflow conditions during ACE-Asia, J. Geophys. Res., 109, D24203, doi:10.1029/2004JD004749, 2004.

Kittelson, D.B.: Engines and nano-particles: a review. Journal of Aerosol Science, 29, 575-588, doi:10.1016/S0021-8502(97)10037-4, 1998.

Kulmala, M., Vehkamäki, H., Petäjä, T., Dal Maso, M., Lauri, A., Kerminen, V. M., Birmili, W., and McMurry, P. H.: Formation and growth rates of ultrafine

atmospheric particles: a review of observations, J. Aerosol Sci., 35, 143-176, doi:10.1016/j.jaerosci.2003.10.003, 2004.

Kumar, P., Robins, A., Vardoulakis, S., and Britter, R.: A review of the characteristics of nanoparticles in the urban atmosphere and the prospects for developing regulatory controls. Atmos. Environ., 44, 5035-5052, doi:10.1016/j.atmosenv.2010.08.016, 2010.

Meng, H., Zhu, Y., Evans, G. J., and Yao, X.: An approach to investigate new particle formation in the vertical direction on the basis of high time-resolution measurements at ground level and sea level, Atmos. Environ., 102, 366-375, doi:10.1016/j.atmosenv.2014.12.016, 2015.

Quinn, P. K., and Bates, T. S.: The case against climate regulation via oceanic phytoplankton sulphur emissions, Nature, 480, 51-56, doi:10.1038/nature10580, 2011.

Seinfeld, J.H., and Pandis, S.N.: Atmospheric chemistry and physics: from air pollution to climate change, John Wiley & Sons, New York, 2012.

Sipilä, M., Sarnela, N., Jokinen, T., Henschel, H., Junninen, H., Kontkanen, J., Richters, S., Kangasluoma, J., Franchin, A., Peräkylä, O., Rissanen, M., Ehn, M., Vehkamäki, H., Kurten, T., Berndt, T., Petäjä, T., Worsnop, D., Ceburnis, D., Kerminen, V.-M., Kulmala, M., O'Dowd, C.: Molecular-scale evidence of aerosol particle formation via sequential addition of HIO3. Nature, 537(7621), 532-534, doi:10.1038/nature19314, 2016.

---

## Author Comment (AC2) · 5 Nov 2018

Overall comment:

This study includes the observations of nanoparticles in six cruises over the marginal seas of China and one cruise to the Northwest Pacific Ocean. The particle number concentration, size distribution, formation rate and growth rate of new particles are discussed. The authors also try to illustrate the roles of anthropogenic and marine biogenic emissions in new particle formation, through analyses on several specific NPF events. The experiments are interesting, and should be beneficial to advance the knowledge on the impacts of human being activities on NPF and global climate

change. However, the experimental design has obvious drawbacks in considering the adequate data to support the analyses in this meaningful research. Nearly no data of the precursors of condensable vapors are available. Though some chemicals, such as the amines and the oxalic acid, in the size-segregated are analyzed, the sampling period even missed the NPF periods, which led to the inappropriateness of using these data to infer the processes and chemical species dominating NPF. I also have serious concern on the explanations to the different relationships between the formation rate and the net maximum increase in the nucleation mode particle number concentration. Similarly, the conclusion that the NPFs, regardless of which categories, are regional phenomenon cannot convince me, since no solid evidence has been provided. In view of the inadequate discussions, misleading inferences and even wrong interpretations, the paper needs to be revised substantially before being considered to be accepted. Specific comments are also given for the authors' reference.

Response: The authors thank the reviewer's comments. We agree that we have no data of the precursors of condensable vapors. The weakness will be added in the revision. The weakness is quietly common in NPF studies in the literature. We also add the results of condensation sinks to support our analysis in revision. A few more comments are also very constructive for us to improve the quality of this manuscript because the related parts in the origin version are indeed misleading or even wrong. We make a substantial revision accordingly and explain why.

For parts of reviewer's comments, we believe that more clarifications are needed to make the analysis more readable. We revise these parts accordingly and explain why. Moreover, the authors may disagree with a few reviewer's comments. We explain why in this response below.

Specific comments:

1. Page 3, "In November 2012, the NO2 column densities were higher in the eastern mainland of China due to the house-heating". House heating is not the sole cause of

elevated NO2 in autumn.

Response: Agree. In addition to the house heating, poor dispersion conditions (e.g. temperature inversion) and other factors may also lead to the elevated column densities of NO2 in November. In revision, it will be revised as "In November 2012, the elevated NO2 column densities in the eastern mainland of China were likely due to combined factors such as intensive house heating, poor dispersion conditions, etc."

2. Page 6, lines 6-10. How do you confirm that these NPF events were the regional NPF events, rather than the local ones that occasionally occurred on the same days? Is there any evidence proving that the air masses were homogeneous on these days, except for the backward trajectories? Since the ship location and the coastal sites were generally in an area influenced by the same monsoon, they always received air masses from the same directions. However, it does not mean that the regional air overrode the properties of local air masses.

Response: We are sorry that we cannot agree with the comment. The authors believe that the reviewer may mix up a few concepts.

In the recent highly cited article entitled as "Measurement of the nucleation of atmospheric aerosol particles", regional NPF events refer to these events occurring in a spatial extent varies from tens to thousands of kilometers (Kulmala et al., 2012, Nat. Protoc.). The reviewer may mix up concepts such as "regional NPF events", "simultaneous NPF events" and "regional-identical NPF events" (Hussein et al. et al., 2009, Atmos. Chem. Phys.). The authors believe that the reviewer was arguing against "regional-identical NPF events" rather than "regional NPF events". The regional-identical NPF events are a subset of simultaneous NPF events. The same can be said for simultaneous NPF events against regional NPF events. The authors also have a big concern for occurring regional-identical NPF events in the marine atmosphere over the marginal seas of China and in the continental atmosphere over the eastern part of China. NPF events usually occur in either less polluted or clean atmospheres. Under such condition, it is hard for the authors to believe that the regional air mass always overwhelms the local air mass in the atmosphere over a large spatial area in the eastern part of China and downwind seas.

The authors agree that measurements at two or even more fixed sites are really difficult to justify NPF events as regional. In fact, mobile measurements over a large spatial scale are well suitable to examine regional NPF events. We will clarify that the on-board observations were made mostly on traveling instead of anchoring at the fixed locations. According to the definition above-mentioned, the NPF events observed over the marginal seas have no doubt to be confirmed as regional events except NPF event on 15 May 2014. The solid evidences include 1) the duration of the NPF events exceeded 3 hours in 22 days out of the total 23 days over the marginal seas of China, 2) on-board observations were made mostly during traveling instead of when anchored at fixed locations. The ship travelled at a speed of 18 km/h. A rough calculation of the spatial span is 18 km/h$\times$NPF time in hours for the NPF events over the sea. The NPF event on 15 May 2014 appeared to last for about one hour due to the ship emissions overwhelming the new particles signal after 09:30, this has been clarified in the revision.

Moreover, simultaneous observations of NPF events at the coastal site on the same day further zoom regional NPF events into simultaneous NPF events, i.e., NPF events occurring on a line over dozen of kilometers in a marginal sea plus at an additional coastal site. The simultaneous NPF events are a subset of regional NPF events and the types of NPF events had been claimed based on several sets of measurements over a large spatial range in literatures (Hussein et al., 2009, Atmos. Chem. Phys., Jeong et al., 2010, Atmos. Chem. Phys., Wang et al., 2013, Atmos. Chem. Phys., Shen et al., 2018, Atmos. Chem. Phys.).

3. Page 6, lines 11-13. From the particle number distributions shown in Fig. A1h, i, l, m, I can hardly believe that these are the regional NPF events. Besides, could the delay be caused by the different weather conditions, or downward transport of nanoparticles

in the afternoon?

Response: On 17 Nov. 2012, the NPF event lasted for 4 hours over the marginal seas and 3 hours at the OUC site (due to the instrument maintenance after 15:00). Even longer duration for NPF events occurred on 7 Nov. 2013. Referred to our response to Comment 2, the two NPF events should be considered as regional events.

We agree that different weather conditions and downward transport of nanoparticles could be ones of causes for the delay of the NPF observed in the coastal atmosphere. Weather conditions can affect concentrations of precursor vapors and affect the occurrence of NPF. In the revision, the sentence will be revised as "Many other factors, such as weather conditions which can affect the concentrations of precursor vapors and gas-aerosol partitioning, downward transport of nanoparticles, etc., might also contribute to the delay."

4. Section 3.2. The observational particle number distributions at OUC were not well presented.

Response: This study focuses on NPF events in marine atmospheres. The authors prefer to revise the title of Section 3.2 as: "Particle number concentrations and size distributions in presence of NPF events against the background in marine atmospheres". As a comparison, we agree that a short summary of particle number size distribution in the coastal atmosphere should be included. For Category 1, the authors will add a short summary: "In the Category 1 data observed at the OUC site, the size distribution of the average particle number concentration was similar to that in the atmosphere over the marginal seas of China. For example, the Aitken mode and accumulation mode particles accounted for approximately 80% of the total particle number concentration. However, the average particle number concentration of $1.4\pm0.8\times104$ particles cm-3 observed at the OUC site increased by one-fold compared to that over the marginal seas of China and by approximately four-fold compared to that over the NWPO."

For Category 2, more information of particle number size distributions at OUC will be

added: "Over the NWPO, the increase in concentration of newly formed particles was limited to particles with diameter lower than 30 nm, possibly because of the growth pathways of newly formed particles being different from those at the OUC site and over the marginal seas, where newly formed particles can grow to diameters up to 60 nm." "For example, Fig. A1a, b showed ceilings of approximately 50 nm, and Fig. A1c, d showed ceilings of approximately 20 nm during the events over the marginal seas and at the OUC site."

5. Page 6, lines 20-23. How did you remove the influence of ship-self emissions? This needs to be demonstrated in methodology.

Response: The ship-emitted particles can be clearly identified in the high-time resolution measurements. First, ship-emitted particles exhibit a uni-modal size distribution at 10-60 nm with a peak at 20-30 nm. There is only a small variation in the particle number size distribution, depending on weather conditions. Second, the number concentration of the ship-emitted particles is an order of magnitude higher than that of the background particles as well as new particles. Third, there are dozens to hundreds of spikes in the particle number concentration when the ship-emitted particle signal is detected. For example, Fig. 1 at the end of this response showed the ship plumes (from 12:32 to 13:01, from 14:57 to 15:37 and from 17:33 to 18:10) with the high particle number concentration ($7.3 \times 10^4 \pm 2.5 \times 10^4$ cm-3). When we do the calculation for FR, GR and NMINP, the three features above-mentioned were used to remove the ship emission periods. We will add the part in the revision.

6. Page 7, lines 13-14. "The increase likely induced by the long-range transport of air pollutants from the continents, inferred from the doubled number concentrations of accumulation mode particles in Category 2 relative to Category 1." This is contradictory to the previous statement that "the concentration increase was limited to particles with the diameter less than 20 nm".

Response: What we exactly want to say are different from those shown in the original

version. Thank for the comment help us realize this. In the revision, we rewrite the part. It is "Compared to Category 1, NPF events greatly enhance the total particle number concentrations (Fig. 2, solid lines) in Category 2 over three regions including the NWPO, the marginal seas of China and OUC, mostly because of a large increase in the number concentration of newly formed particles. Over the NWPO, the concentration increase of newly formed particles was limited to particles with diameters lower than 30 nm, possibly because of the growth pathways of newly formed particles being different from those at the OUC site and over the marginal seas, where newly formed particles can grow to diameters up to 60 nm. "

7. Page 7, lines 19-22. The authors should illustrate in more details the size ceilings that the particles could grow up to. What caused the different ceilings, and what were the implications from the differences in particle size distributions?

Response: Agree. The part will be revised as "The results were caused by varying size ceilings in the growth of newly formed particles, i.e., the growth of newly formed particles apparently stopped when they grew to the maximum sizes during these events. For example, Fig. A1a, b show ceilings of approximately 50 nm, and Fig. A1c, d show ceilings of approximately 20 nm during the events over the marginal seas and at the OUC site. In fact, a size ceiling is a common phenomenon during NPF events occurring in various urban or coastal atmospheres, as highlighted by Zhu et al. (2014, 2017) and Man et al. (2015). They also proposed that the size ceiling is associated with the thermodynamic partitioning of semi-volatile species in growing newly formed particles."

We also agree with this reviewer, i.e., it is important to ask "What caused the different ceilings, and what were the implications from the differences in particle size distributions". Theoretically, which semi-volatile species dominate the growth of newly formed particles and what their vapor concentrations are in the atmosphere during NPF events are critical to fully answer the question. In absence of the two results, we cannot speculate more from the differences in particle size distributions. Honestly, we have no breakthrough progress on determining these semi-volatile species in the last decade.

However, we believe that the ceiling phenomenon would stimulate more future studies in research community.

8. Page 8, lines 10-15. Condensation sink is an important factor influencing particle formation. Throughout the manuscript, CS has never been presented and has seldom referenced for discussions. The lack of measurements of condensable vapors makes so many inferences in the paper not reliable, not to say some inferences are contradictory to common sense. For example, here I cannot believe that the loadings of precursors favorable for the formation of new particles were higher over the marginal seas than in the coastal area. Evidences need to be provided to support the inferences.

Response: The authors agree that it is worthy of the inclusion of condensation sink. Condensation sink prior to or during NPF event plays an important role in removing condensation vapors, although it may or may not dominantly determine concentrations of condensation vapors. In this study, the CS over the marginal seas were 1.1±1.0 (10-2 s-1), and much lower than that at OUC site of 4.1±2.0 (10-2 s-1) during simultaneous NPF events. However, no significant negative correlation between the FR/GR and CS was observed (Fig. 2 at the end of this response). The results will be added in the revision and Supporting Information (Fig. S4 in revision).

In addition, it is hard to say that the cleaner atmosphere should have fewer loadings of precursors, e.g., the FR is lower than that in the clean atmosphere than in polluted atmospheres. In our previous studies to compare NPF events in the atmospheres at different pollution levels (Qingdao, Hong Kong, Toronto), we did not find a clear relationship between the degree of air pollution and FR (Zhu et al., 2014, Man et al., 2015). So does in a number of investigations summarized by Kulmala et al., 2004. Kulmala et al. (2005) claimed that the larger CS in the polluted atmospheres can be compensated by a larger vapor source rate, which can up to four orders of magnitude larger than in the clean atmospheres.

Larger FR should be an important evidence for higher concentrations of condensation

vapors based on the nucleation theory. We agree that the measurement of condensation vapors are more direct evidences than FR. The weakness of lack of measurements of condensation vapors will be added in the revision.

9. Page 8, lines 17-32. I cannot understand why the higher formation rate did not result in larger increase of nucleation mode particles, note that the formation rate is closely related to the increase of nucleation mode particles if looking at the calculation formula of formation rate. All the explanations are based on the assumptions, which cannot convince me. The authors should provide more evidences to validate their assumptions. The authors state that "the NMINP was always determined by the consumed $H_2SO_4$ vapor for nucleation". Sorry for that I cannot accept this view. How about the number of nucleation mode particles when the organic vapors facilitated the nucleation and particle growth to the detectable size? The so called threshold of formation rate, i.e. 8 cm-3s-1, was exactly the same as that reported in the study previously published by the same authors. This cannot convince the readers unless the similar phenomenon has been reported by other groups. I tried to understand the authors' view by finding the clues from the paper "Simultaneous measurements of new particle formation at 1 s time resolution at a street site and a rooftop site". However, it is hard for me to follow up the authors in many points. For example, in this paper, the sentence "Supposing that sulfuric acid vapors are completely nucleated, followed by the nucleated particles growing to the detectable size, the yields of newly formed particles are determined mainly by the supply of sulfuric acid vapor and are less affected by the formation rate" is problematic. How could you separate the role of sulfuric acid from the formation rate, as sulfuric acid plays critical role in nucleation? In the sentence "Scenario 1: $H_2SO_4$ vapor is relatively sufficient against NucOrg, and J8 is therefore determined mainly by the availability of NucOrg vapor. A good correlation is theoretically expected for J8 and NMINP". To be honest, I do not understand the logics behind.

Response: In the revision, we add more clarification to better defense our arguments. We also try our best to explain the difference between our analysis and the reviewer's

thoughts.

We assume that NPF rapidly stops after dozens of minutes bursting. This is consistent with huge measurements of banana-shaped NPF events reported in literature (Kulmala et al., 2004). If NPF continuously occurs during the whole event, a fan-shaped NPF event would be detected instead of banana-shaped NPF event. This is because of continuous formation and growth of new particles. In fact, a fan-shaped NPF event was hardly observed. Moreover, in Fig. 1 published by Yue et al, 2010 (Atmos. Environ.) and Fig. 3-4 published by Wang et al., 2011 (Atmos. Chem. Phys.), NPF rapidly stops after dozens of minutes bursting with rapid consumption of H2SO4 vapor. These studies also directly supports our assumption. We will add the part of analysis in the revision.

The reviewer commented "note that the formation rate is closely related to the increase of nucleation mode particles if looking at the calculation formula of formation rate." The comment does not sound scientific. This is no doubt that FR is determined mainly by the nucleation mechanism (i.e., nucleation of sulfuric acid vapor enhanced by organics). The equation is used to measure the apparent formation rate and has nothing to do with nucleation mechanisms. Technically, we can measure the vehicle speed on basis of vehicle traveling mileage in a fixed time. We clearly know that a vehicle speed depend mainly on engines and fuels, etc., but has nothing to do with vehicle traveling mileage. Moreover, the largest mileage of a vehicle is mainly determined by the used liters of fuel in vehicle tank. Engines and other factors can greatly affect vehicle speed, but the influence on the largest traveling mileage is not comparable to that of the liters of fuel in vehicle tank. The same can be said for FR and NMINP. In NPF events, the authors technically consider sulfuric acid vapor as fuel while organic as engine and other factors affecting vehicle speeds.

The reviewer commented "the authors state that "the NMINP was always determined by the consumed H2SO4 vapor for nucleation". Sorry for that I cannot accept this view. How about the number of nucleation mode particles when the organic vapors facilitated

the nucleation and particle growth to the detectable size?" The authors may disagree with the comments. Regarded much low nucleation rates of inorganic vapors reported so far, the authors strongly believe that all NPF events observed in the atmospheric boundary layer on the earth were facilitated by organic vapors to some extent. When the organic vapors don't facilitate the nucleation and particle growth to the detectable size, there are no NPF events to be observed in the atmospheric boundary layer and the NMIMP is zero.

We also agree that the threshold of formation rate, i.e. 8 cm-3s-1, may be coincidentally consistent with our previous study. This needs more work to be confirmed. The part will be added in the revision.

10. Page 9, lines 29-33. The concurrent occurrences of class II NPF events at the coastal site and over the marginal seas could not be an evidence of the regional characteristics. The particle number distributions at the two sites were quite different on the days specified by the authors (Figure A1). Besides, it is difficult to convince me with the backward trajectories. The two sites were in a same region under the influence of the same monsoon. Even so, the air masses could be totally different in chemical compositions when they passed over the different cities. With no chemical information or mesoscale simulation, it is hard to say the two sites were interacted and the regional NPF events occurred at the two sites.

Response: Referred to our response to Comment 2, the reviewer may mix up a few concepts. The class II NPF events lasted for 3-5 hours and these NPF events should be considered as regional events.

11. Page 10, lines 1-6. Condensable vapors are of course critical in NPF. However, it is not reasonable to simply attribute the different characteristics of NPF to the abundances of the condensable vapors. Other factors, such as the preexisting particles and the meteorological conditions also influence the NPF. In this case, more preexisting particles with larger diameters existed at the marginal sea site. Could this also account

for the insignificant particle growth?

Response: We agree that weather conditions can affect the growth of newly formed particles by changing gas-aerosol partitioning. We will revise the part accordingly: "Theoretically, higher CS can remove more condensable vapors and consequently reduce the vapor pressure of precursors. In this case, the apparent particle growth was undetectable in the marine atmosphere with the smaller CS of $0.6\pm0.1$ ($10^{-2}$ s-1) against the value of $5.3\pm0.2$ ($10^{-2}$ s-1) in the coastal atmosphere. However, the apparent growth of new particles observed at the OUC site indicates that 1) the concentrations of condensable vapors are higher than the required value to support the growth; 2) CS is not the dominant factor to determine the growth. Apart from the condensation vapor, weather conditions can also affect the growth of newly formed particles by changing gas-aerosol partitioning."

We may disagree with other parts of reviewer's comments. Preexisting particles can remove condensable vapors, nucleating clusters and newly formed particles from the atmosphere and then affect NPF and the growth of new particles. In addition to affecting condensable vapors, the authors cannot figure out other pathways for preexisting particles to affect particle growing larger than 10 nm.

12. Section 4.1. I do not agree that new particle formation occurred in this case, i.e. 30 August 2015.

Response: The NPF event on 30 August 2015 followed the definition proposed by Dal Maso et al. (2005), Hirsikko et al. (2007) and Kulmala et al. (2012), i.e., the nucleation mode of newly formed particle was observed for about 6 hours, and newly formed particles grew up to approximately 20 nm. In the revision, the time series of N<30 nm and CS (Fig. 3 at the end of this response) will be added and discussed. "To delve into the characteristics and evidence of oceanic precursors related NPF event on 30 August 2015, the transport pathway on that day was first zoomed in Fig. 6a. As is illustrated in Fig. 6b, the NPF event started to be observed at 09:40 under meteorological conditions

with ambient temperature of 26°C, high relative humidity of 74%, and low wind speed of 1.5 m s-1 (not shown). During the first hour, the N<30nm increased from 0.6×103 cm-3 to 1.7×103 cm-3. The weaker NPF was associated with higher CS (2×10-2 s-1). When CS decreasing to approximately 1×10-2 s-1 after 11:00, the N<30nm sharply increased to 3×103 cm-3, and Dpg increase from 13 nm to 18 nm during the following 3 hours with the growth rate of 1.7 nm h-1. The signal of new particles disappeared at approximately 16:00. The overall NMINP was 5-20 times lower than all the other NPF events over the marginal seas, and the overall FR of 0.3 cm-3 s-1 was the minimum in this study."

We are sorry for color bar used in Fig. 6b (original version), which may mislead the reviewer. To make the weak signal of new particles to be visible, the scale of color bar in Fig. 6b was one fifth of other contour figures. The choice also makes the signal of pre-existing particles darker in Fig 6b in comparison with other contour figures.

13. Page 11, lines 12-15. Figure 7c does not show the altitude variation of the backward trajectories.

Response: We will add the altitudes in the supplementary as Fig. S7 (as shown in Fig. 4 at the end of this response).

14. Page 11, lines 16-26. The sampling periods of MOUDI samples were after the NPF events, not including the hours when the new particles were formed and grew up. I would doubt the reasonability of using these data to infer the chemical species dominating NPF. Same for the other similar discussions.

Response: We thank the comment. In the original version, the inclusion of MOUDI data on those two days are not well justified. The analysis is also too speculative to be convincing. The part will be revised as below:

"One set of MOUDI samples was collected during the period from 11:12 to 23:33. Although the sampling period had several hours delay against the NPF period on that

day, the air mass back trajectories swept the oceanic zone were highly consistent between the two periods (Fig. S8, as shown in Fig. 5 at the end of this response). The concentrations of particulate chemical species were thereby used to argue the polluted extent of air mass at these periods. The mass concentration of nss-sulfate and oxalate in particles less than 10 $\mu$m was 1.9 $\mu$g m-3 and 0.12 $\mu$g m-3 (derived from Fig. 7d), respectively, higher than in other non-NPF days in this study. Previous studies, e.g., Mukai et al., (1995), Matsumoto et al. (1997), and Jung et al. (2014), reported the mass concentration of nss-sulfate was approximately 0.5 $\mu$g m-3 in the clean background over the NWPO. The elevated concentration of nss-sulfate and oxalate on 8 April suggested the enhanced anthropogenic precursors input which was very likely from the continent of Japan based on the calculated air mass back trajectories (Fig. 7c,d). The MOUDI's data implied that the NPF event likely occurred in the air masses rich in anthropogenic precursors.

Compared to the event above on 8 April, the event on 13 April showed a longer NPF duration, i.e., the NPF event lasted from 07:50 to approximately 08:50 (Fig. 7b). The new particles signal was intermittently observed and the FR was difficult to calculate. The total particle number concentrations increased from 0.3×104 cm-3 to the maximum of 2.6×104 cm-3 during the NPF event, and the NMINP was 1.4×104 cm-3. The Dpg increased from 8 nm to 14 nm in one hour, and the estimated GR was 3.6 nm h-1. One set of MOUDI samples was collected immediately after the event during the period from 09:10 to 21:05. Again, the calculated air mass back trajectories were consistent between the NPF period and the MOUDI's sampling period (Fig. S8, as shown in Fig. 5 at the end of this response). The mass concentration of nss-sulfate and oxalate in particles less than 10 $\mu$m was only 0.6 $\mu$g m-3 and 0.05 $\mu$g m-3. The values were close to the clean background of NWPO, indicating a much low anthropogenic input on 13 April (Fig. 7e). It is interesting that the NMINP was similar to each other during the two NPF events, although the air mass on 8 April was slightly polluted by anthropogenic inputs. However, due to lack of the measurements of precursor vapors, what caused NPF events needs further study."

Interactive
comment

15. Page 12, lines 8-10. I do not understand the logics behind this inference, though it is true that the AR increased after Dpg was higher than 50 nm. Why not present the number concentration of >50 nm particles or its fraction in total particles against the NCCN? It would be a more direct way to link the particles larger than 50 nm to CCN.

Response: "threshold" indeed causes misleading. Not all particles larger than 50 nm can be activated as CCN. In the revision, the part has been revised as "At SS of 0.4%, the Dpg increased from 19 nm to 50 nm during 10:40-13:10 (black circles in Fig. 8a) with AR fluctuating at 0.1-0.2 (Fig. 8c). After 13:10, the Dpg increased from 50 nm to 77 nm with increasing AR from ∼0.2 to ∼0.4. The results are consistent with those reported in the literature, i.e., particles smaller than 50 nm are unlikely activated as CCN at SS=0.4% (Dusek et al., 2006; Petters and Kreidenweis, 2007).

Following the reviewer's comments, we plotted time series of the number concentration of >50 nm particles (N>50nm) and the NCCN (as shown in Fig. 6 at the end of this response). Variations between N>50 nm and AR are clearly inconsistent. For example, AR showed an increasing trend from 0.2 to 0.4 during 13:10-15:00. N>50 nm decreased from $1.8 \times 10^4$ cm-3 to $1.1 \times 10^4$ cm-3 during the period of 13:10-13:30, then increasing to $2.4 \times 10^4$ cm-3 at 14:30, followed a decreasing trend after 14:30. Atmospheric particles with the diameter larger than 50 nm include not only the grown new particles, but also preexisting particles. The inconsistency is not very surprised. The reviewer's comment is valid only when the number concentration of preexisting particles >50 nm was either near constant or was negligible relative to grown new particles during the growth period. This is not the fact. Therefore, we disagree with the reviewer on this point.

16. Caption of Figure 3, what does "exteriors" mean? Why should they be excluded from the regression? Figure 4, what does the black dots represent, same for the other figures? Figure 9, what does the highlighted area denote for?

Response: It should be outlier rather than exterior. We are sorry for our language

problem. In Figure 3a, there was a moderately good linear correlation at FRs $\leq$ 8 cm-3 s-1. The data points with FRs larger than 10 cm-3 s-1 are deviated largely from the regression curve obtained from the data with FRs $\leq$ 8 cm-3 s-1 and are thereby treated as outliers. For example, in the linear regression question of [NMINP]=3.9$\times$103$\times$FR, r=0.83 P<0.01, we consider three times of standard deviation for the slope. At the FR of 11.8 cm-3 s-1, the range of NMINP is predicted from 3.56$\times$104 to 5.64$\times$104 particles cm-3. The observed NMINP was only 1.17$\times$104 particles cm-3 and largely deviated from the range. In Figure 3b, the black triangle represents the GR of 26.3 nm h-1 and also deviated largely from the regression curve obtained from other data. The point is also treated as an outlier. This will be added in the figure caption.

The black dots in the contour plot of NPF events (Fig. 4, Fig. 6b, Fig. 7a, b, Fig. 8a and Fig. A1 in original version) represent the fitted geometric median diameter of new particles (Dpg) in 1-minute time resolution. The clarification will be added in the revised caption of Fig. 4.

In the revision, the shading in Fig 9 will be removed to avoid any misleading.

17. The manuscript needs to be grammatically checked by an editing company or a native English speaker professor.

Response: Thanks. The revised version will be language-edited.

Reference:

[revised manuscript text omitted]

---

## Author Response (AR1)

*This manuscript combines measurements conducted over several marine cruises to investigate atmospheric new particle formation (NPF) and growth in the marine atmosphere. The paper appears scientifically sound and original enough to merit publications. In its current form, the paper requires, however, important revisions, especially what it comes to the technical quality of the paper.*

**Response:** The authors thank the reviewer's comments and try our best to respond and revise our manuscript accordingly. Note that there is a minor language-editing change in this version from on-line version.

*Scientific issues*

*The second paragraph of Introduction gives a background on NPF in the marine atmosphere. It contains a sentences discussing the role of amines in NPF (lines 22-23 on page 2) which is no way related to marine NPF. I recommend this sentence to be removed from here. The discussion on role of ions in coastal NPF does not include the paper by Sipila et al (2016, Nature) that gives the most detailed molecular view on this process published so far.*

**Response:** Ocean is one of the important sources of atmospheric amines. The authors will revise the sentence to "Moreover, amines, which can be produced through excretion and metabolism by a variety of marine organisms, were reported to enhance $H_2SO_4$-$H_2O$ nucleation and promote the growth of newly formed particles". The reference of Sipilä et al. (2016) will be added in the revised manuscript.

*Page 6, line 9: the numbers appear too accurate. I suggest writing: ...event to be at least 50-500 km.*

**Response:** We corrected the sentence accordingly.

*Page 6, line 26: the selected border between the Aitken and accumulation mode (50 nm) is very untypical. Normally in a scientific literature, it is assumed to be between 80 and 100 nm. Please correct or give a reason for this choice.*

**Response:** Agree. On basis of those highly cited references, e.g., Kittelson, 1998, Kulmala et al., 2004, Kumar, et al., 2010; Seinfeld and Pandis, 2012, we revised the sentence as "The Aitken mode (30-100 nm) and accumulation mode (100-500 nm) were usually overlapped at the size range of 30-500 nm, with a minor nucleation mode at sizes below 30 nm".

*Page 11, lines 7-8: this statement requires a couple of more, and more recent, references.*

**Response:** In revision, the authors added the reference of Buzorius et al. (2004, J. Geophys. Res.), Quinn and Bates (2011, Nature) and Meng et al. (2015, Atmos. Environ.), which reported that new particles can be formed above the marine boundary layer and mixing downward.

*Page 13, lines 12-13: I am confused about this assumption. Do you mean that there should be no sulfate in nm sized particles?*

**Response:** In the revision, we added "Appreciable amount of $SO_4^{2-}$ should exist in <56 nm

particles, but this amount might be much smaller than sampling artifacts."

*Page 14, line 12: dozens of minutes to one hours sounds a very strange range because dozens corresponds to several tens of minutes and one hour is the same (60 min). Please correct or modify.*

**Response:** The authors revised as "from 17 min to one hour" in revision.

*The paper has several sentences that are either difficult to understand or written in bad style, so they need to be rewritten. They are in the following places: page 4, lines 1-2; page 5, lines 30-31; page 7, lines 20-22; page 7, lines 27-31; page 8, lines 23-29; page 9, lines 2-3; page 9, lines 24-26; page 9, lines 29-30; page 10, lines 1-5; page 11, line 4-5; page 11, lines 12-15; page 12, lines 11-13; page 14, lines 10-11; page 10, line 14-15; page 14, lines 22-28.*

**Response:** The authors have rewritten these sentences and double-checked the language using a profession service.

*Technical issues*

*The paper refers to figures and tables marked as S1, S2 etc. They are in Appendix, so A1, A2 etc would be more logical way to refer to them.*

**Response:** The original paper contain the Appendix in text and the Supplementary as attachment. Figure and table in appendix marked as Fig. A1 and Table A1, and the figures and tables in supplementary marked as Fig. S1 and table S1. We have double check it.

*The following grammatical corrections are needed (the text below give the correct way to write them):*

*page 3, line 32:…monsuun prevails…*

*page 6, line 10:…sinks are two…*

*page 6, line 20: …first classified…*

*page 6, line 26: We first discuss category I data over the marginal…*

*page 7, line 1: …lower than that over the marginal…*

*page 7, line 2 :…as over the marginal seas (20%), indicating…*

*page 7, line 5: …altitudes*

*page 7, line 9:…with diameters lower than 20 nm*

*page 7, line 11: …higher than that reported in previous…*

*page 7, line 16: a comparable*

*page 7, line 27: over the…during the three…*

*page 8, line 1:…with a high…*

*page 8, lines 2-4: ...intermittent occurrence of nucleation...here were much higher than those observed in previous...Altogether, considering both...*

*page 8, line 6: we next compare...*

*page 8, line 8:...larger mean values*

*page 8, line 9:...whereas comparable...*

*page 8, line 10: over the marginal...*

*page 8, lines 12-13: precursors, such as...vapors, were...*

*page 8, line 17: ...no obvious*

*page 9, line 1: ...24 NPF days, except on one day when it was 77 nm. ...could be identified*

*page 9, line 5: ...were able to grow...*

*page 9, line 18:...event occur in regional*

*page 9, lines 19-20:...event are mostly local phenomena reported in a few studies made over...*

*page 9, line 21:...in an urban...*

*page 10, line 11: ...were accompanied*

*page 10, line 16: ...zoomed in (Fig. 6a).*

*page 10, line 17:...high relative humidity of 74% and low wind speed of ...*

*page 10, line 18: ...characterized by a low*

*page 10, line 19: during the first hour*

*page 10, line 21-22: ...during the first 30 minutes...fluctuated...during the following 3 hours...*

*page 10, line 28: ...suggests a strong*

*page 11, line 3: and it lasted...the total particle*

*page 11, line 16: ...compounds may be involved in*

*page 11, line 17: day was analyzed*

*page 11, line 18: particles smaller than 10*

*page 11, line 19:...(derived...respectively, higher than in other*

*page 11, line 22: ...involved in... moderately high*

*page 11, line 31: lower than*

*page 12, line 6: ...at an initially high relative humidity of*

*page 12, line 17: we found that the*

*page 12, line 25-26: implying that the majority of... particle were able to grow to CCN at*

*page 12, line 28: to act as CCN*

*page 13, line 5: to play*

*page 13, line 12: errors in*

*page 14, line 18: Moderately good…were obtained*

*Finally, please check out carefully the language of the abstract*

**Response:** The authors thank the reviewer's grammatical comments. We corrected all of the errors above and double-checked the language using a profession service throughout the manuscript.

*Anonymous Referee #2*

*Overall comment:*

*This study includes the observations of nanoparticles in six cruises over the marginal seas of China and one cruise to the Northwest Pacific Ocean. The particle number concentration, size distribution, formation rate and growth rate of new particles are discussed. The authors also try to illustrate the roles of anthropogenic and marine biogenic emissions in new particle formation, through analyses on several specific NPF events. The experiments are interesting, and should be beneficial to advance the knowledge on the impacts of human being activities on NPF and global climate change. However, the experimental design has obvious drawbacks in considering the adequate data to support the analyses in this meaningful research. Nearly no data of the precursors of condensable vapors are available. Though some chemicals, such as the amines and the oxalic acid, in the size-segregated are analyzed, the sampling period even missed the NPF periods, which led to the inappropriateness of using these data to infer the processes and chemical species dominating NPF. I also have serious concern on the explanations to the different relationships between the formation rate and the net maximum increase in the nucleation mode particle number concentration. Similarly, the conclusion that the NPFs, regardless of which categories, are regional phenomenon cannot convince me, since no solid evidence has been provided. In view of the inadequate discussions, misleading inferences and even wrong interpretations, the paper needs to be revised substantially before being considered to be accepted. Specific comments are also given for the authors' reference.*

**Response:** The authors thank the reviewer's comments. We agree that we have no data of the precursors of condensable vapors. The weakness will be added in the revision. The weakness is quietly common in NPF studies in the literature. We also add the results of condensation sinks to support our analysis in revision. A few more comments are also very constructive for us to improve the quality of this manuscript because the related parts in the origin version are indeed misleading or even wrong. We make a substantial revision accordingly and explain why.

For parts of reviewer's comments, we believe that more clarifications are needed to make the analysis more readable. We revise these parts accordingly and explain why. Moreover, the authors may disagree with a few reviewer's comments. We explain why in this response below.

Note that there is a minor language-editing change in this version from on-line version.

*Specific comments:*

*1. Page 3, "In November 2012, the NO2 column densities were higher in the eastern mainland of China due to the house-heating". House heating is not the sole cause of elevated NO2 in autumn.*

**Response:** Agree. In addition to the house heating, poor dispersion conditions (e.g. temperature inversion) and other factors may also lead to the elevated column densities of $NO_2$ in November. In revision, it will be revised as "In November 2012, the elevated NO2

column densities in the eastern mainland of China were likely caused by multiple factors, such as intensive house heating and poorer dispersive conditions."

*2. Page 6, lines 6-10. How do you confirm that these NPF events were the regional NPF events, rather than the local ones that occasionally occurred on the same days? Is there any evidence proving that the air masses were homogeneous on these days, except for the backward trajectories? Since the ship location and the coastal sites were generally in an area influenced by the same monsoon, they always received air masses from the same directions. However, it does not mean that the regional air overrode the properties of local air masses.*

**Response:** We are sorry that we cannot agree with the comment. The authors believe that the reviewer may mix up a few concepts.

In the recent highly cited article entitled as "Measurement of the nucleation of atmospheric aerosol particles", regional NPF events refer to these events occurring in a spatial extent varies from tens to thousands of kilometers (Kulmala et al., 2012, Nat. Protoc.). The reviewer may mix up concepts such as "regional NPF events", "simultaneous NPF events" and "regional-identical NPF events" (Hussein et al. et al., 2009, Atmos. Chem. Phys.). The authors believe that the reviewer was arguing against "regional-identical NPF events" rather than "regional NPF events". The regional-identical NPF events are a subset of simultaneous NPF events. The same can be said for simultaneous NPF events against regional NPF events. The authors also have a big concern for occurring regional-identical NPF events in the marine atmosphere over the marginal seas of China and in the continental atmosphere over the eastern part of China. NPF events usually occur in either less polluted or clean atmospheres. Under such condition, it is hard for the authors to believe that the regional air mass always overwhelms the local air mass in the atmosphere over a large spatial area in the eastern part of China and downwind seas.

The authors agree that measurements at two or even more fixed sites are really difficult to justify NPF events as regional. In fact, mobile measurements over a large spatial scale are well suitable to examine regional NPF events. We will clarify that the on-board observations were made mostly on traveling instead of anchoring at the fixed locations. According to the definition above-mentioned, the NPF events observed over the marginal seas have no doubt to be confirmed as regional events except for the NPF event on 15 May 2014. The solid evidences include 1) the duration of the NPF events exceeded 3 hr on 22 days out of the total of 23 days over the marginal seas of China, 2) on-board observations were made mostly during traveling instead of when anchored at fixed locations. The ship traveled at a speed of 18 km/h. A rough calculation of the spatial span is 18 km/h×NPF time in hours for the NPF events over the sea. The NPF event on 15 May 2014 appeared to last for about one hour because the ship emissions overwhelmed the new particle signal after 09:30, this has been clarified in the revision.

Moreover, simultaneous observations of NPF events at the coastal site on the same day further zoom regional NPF events into simultaneous NPF events, i.e., NPF events occurring on a line over dozen of kilometers in the marginal sea plus at the additional coastal site. The simultaneous NPF events are a subset of regional NPF events, and the types of NPF events had been claimed based on several sets of measurements over a large spatial range in the

literature (Hussein et al., 2009, Atmos. Chem. Phys., Jeong et al., 2010, Atmos. Chem. Phys., Wang et al., 2013, Atmos. Chem. Phys., Shen et al., 2018, Atmos. Chem. Phys.)

*3. Page 6, lines 11-13. From the particle number distributions shown in Fig. A1h, i, l, m, I can hardly believe that these are the regional NPF events. Besides, could the delay be caused by the different weather conditions, or downward transport of nanoparticles in the afternoon?*

**Response:** On 17 Nov. 2012, the NPF event lasted for 4 hours over the marginal seas and 3 hours at the OUC site (due to the instrument maintenance after 15:00). Even longer duration for NPF events occurred on 7 Nov. 2013. Referred to our response to Comment 2, the two NPF events should be considered as regional events.

We agree that different weather conditions and downward transport of nanoparticles could be ones of causes for the delay of the NPF observed in the coastal atmosphere. Weather conditions can affect concentrations of precursor vapors and affect the occurrence of NPF. In the revision, the sentence will be revised as "Many other factors, such as weather conditions which can affect the concentrations of precursor vapors and gas-aerosol partitioning, and the downward transport of nanoparticles, etc., might also contribute to such delays."

*4. Section 3.2. The observational particle number distributions at OUC were not well presented.*

**Response:** This study focuses on NPF events in marine atmospheres. The authors prefer to revise the title of section 3.2 as: "Particle number concentrations and size distributions in the presence of NPF events against the background in marine atmospheres". As a comparison, we agree that a short summary of particle number size distribution in the coastal atmosphere should be included. For Category 1, the authors will add a short summary: "In the Category 1 data from the OUC site, the size distribution of the average particle number concentration was similar to that in the atmosphere over the marginal seas of China. For example, the Aitken mode and accumulation mode particles accounted for approximately 80% of the total particle number concentration. However, the average particle number concentration of $1.4\pm0.8\times10^4$ particles cm$^{-3}$ at the OUC site increased by one fold compared to that over the marginal seas of China and approximately four fold compared to that over the NWPO. "

For Category 2, more information of particle number size distributions at OUC will be added: "Over the NWPO, the increase in concentration of newly formed particles was limited to particles with diameters lower than 30 nm, possibly because the growth pathways of newly formed particles were different from those at the OUC site and over the marginal seas, where newly formed particles could grow to diameters of 60 nm." "For example, Fig. A1a, b show ceilings of approximately 50 nm, and Fig. A1c, d show ceilings of approximately 20 nm during the events over the marginal seas and at the OUC site. "

*5. Page 6, lines 20-23. How did you remove the influence of ship-self emissions? This needs to be demonstrated in methodology.*

**Response:** The ship-emitted particles can be clearly identified in the high-time resolution measurements. First, ship-emitted particles exhibited a uni-modal size distribution at 10-60 nm with a peak at 20-30 nm. Only a small variation existed in the particle number size

distribution, depending on the weather conditions. Second, the number concentration of the ship-emitted particles was an order of magnitude higher than that of the background particles and new particles. Third, dozens to hundreds of spikes were present in the particle number concentration when ship-emitted particle signals were detected. For example, Fig. R1 showed the ship plumes (from 12:32 to 13:01, from 14:57 to 15:37 and from 17:33 to 18:10) with the high particle number concentration ($7.3 \times 10^4 \pm 2.5 \times 10^4$ cm$^{-3}$). When we do the calculation for FR, GR and NMINP, the three features above-mentioned were used to remove the ship emission periods. We will add the part in the revision.

[Figure]

Fig. R1 Size distribution of ship emitted particles, new particles and background particles on 14 Nov. 2012 .

*6. Page 7, lines 13-14. "The increase likely induced by the long-range transport of air pollutants from the continents, inferred from the doubled number concentrations of accumulation mode particles in Category 2 relative to Category 1." This is contradictory to the previous statement that "the concentration increase was limited to particles with the diameter less than 20 nm".*

**Response:** What we exactly want to say are different from those shown in the original version. Thank for the comment help us realize this. In the revision, we rewrite the part. It is "Compared to Category 1, NPF events greatly enhanced the total particle number concentrations (Fig. 2, solid lines) in Category 2 over the NWPO, the marginal seas of China and at the OUC site, mostly because of a large increase in the number concentration of newly formed particles. Over the NWPO, the increase in concentration of newly formed particles was limited to particles with diameters lower than 30 nm, possibly because the growth pathways of newly formed particles were different from those at the OUC site and over the marginal seas, where newly formed particles could grow to diameters of 60 nm. "

*7. Page 7, lines 19-22. The authors should illustrate in more details the size ceilings that the particles could grow up to. What caused the different ceilings, and what were the implications from the differences in particle size distributions?*

**Response:** Agree. The part will be revised as "The results were caused by varying size ceilings in the growth of newly formed particles, i.e., the growth of newly formed particles apparently stopped when they grew to the maximum sizes during these events. For example, Fig. A1a, b show ceilings of approximately 50 nm, and Fig. A1c, d show ceilings of approximately 20 nm during the events over the marginal seas and at the OUC site. In fact, a size ceiling is a common phenomenon during NPF events in various urban or coastal atmospheres, as highlighted by Zhu et al. (2014, 2017) and Man et al. (2015). These authors also proposed that the size ceiling is associated with the thermodynamic partitioning of semi-volatile species in growing newly formed particles. "

We also agree with this reviewer, i.e., it is important to ask "What caused the different ceilings, and what were the implications from the differences in particle size distributions". Theoretically, which semi-volatile species dominate the growth of newly formed particles and what their vapor concentrations are in the atmosphere during NPF events are critical to fully answer the question. In absence of the two results, we cannot speculate more from the differences in particle size distributions. Honestly, we have no breakthrough progress on determining these semi-volatile species in the last decade. However, we believe that the ceiling phenomenon would stimulate more future studies in research community.

*8. Page 8, lines 10-15. Condensation sink is an important factor influencing particle formation. Throughout the manuscript, CS has never been presented and has seldom referenced for discussions. The lack of measurements of condensable vapors makes so many inferences in the paper not reliable, not to say some inferences are contradictory to common sense. For example, here I cannot believe that the loadings of precursors favorable for the formation of new particles were higher over the marginal seas than in the coastal area. Evidences need to be provided to support the inferences.*

**Response:** The authors agree that it is worthy of the inclusion of condensation sink. Condensation sink prior to or during NPF event plays an important role in removing condensation vapors, although it may or may not dominantly determine concentrations of condensation vapors. In this study, the CS over the marginal seas were $1.1\pm1.0$ ($10^{-2}$ s$^{-1}$), and much lower than those at OUC site of $4.1\pm2.0$ ($10^{-2}$ s$^{-1}$) during simultaneous NPF events. However, no significant negative correlation between the FR/GR and CS was observed (Fig. R2). The results will be added in the revision and Supporting Information (add as Fig. S4 in revision).

In addition, it is hard to say that the cleaner atmosphere should have fewer loadings of precursors, e.g., the FR is lower than that in the clean atmosphere than in polluted atmospheres. In our previous studies to compare NPF events in the atmospheres at different pollution levels (Qingdao, Hong Kong, Toronto), we did not find a clear relationship between the degree of air pollution and FR (Zhu et al., 2014, Man et al., 2015). So does in a number of investigations summarized by Kulmala et al., 2004. Kulmala et al. (2005) claimed that the larger CS in the polluted atmospheres can be compensated by a larger vapor source rate,

which can up to four orders of magnitude larger than in the clean atmospheres.

Larger FR should be an important evidence for higher concentrations of condensation vapors based on the nucleation theory. We agree that the measurement of condensation vapors are more direct evidences than FR. The weakness of lack of measurements of condensation vapors will be added in the revision.

[Figure]

Fig. R2 Relationship of the condensation sink (CS) with the formation rate (FR) and growth rate (GR) over the marine (NWPO and marginal seas) and at the OUC site (the solid markers represent the simultaneous NPF events).

*9. Page 8, lines 17-32. I cannot understand why the higher formation rate did not result in larger increase of nucleation mode particles, note that the formation rate is closely related to the increase of nucleation mode particles if looking at the calculation formula of formation rate. All the explanations are based on the assumptions, which cannot convince me. The authors should provide more evidences to validate their assumptions. The authors state that "the NMINP was always determined by the consumed H2SO4 vapor for nucleation". Sorry for that I cannot accept this view. How about the number of nucleation mode particles when the organic vapors facilitated the nucleation and particle growth to the detectable size? The so called threshold of formation rate, i.e. 8 cm-3s-1, was exactly the same as that reported in the study previously published by the same authors. This cannot convince the readers unless the similar phenomenon has been reported by other groups. I tried to understand the authors' view by finding the clues from the paper "Simultaneous measurements of new particle formation at 1 s time resolution at a street site and a rooftop site". However, it is hard for me to follow up the authors in many points. For example, in this paper, the sentence "Supposing that sulfuric acid vapors are completely nucleated, followed by the nucleated particles*

*growing to the detectable size, the yields of newly formed particles are determined mainly by the supply of sulfuric acid vapor and are less affected by the formation rate" is problematic. How could you separate the role of sulfuric acid from the formation rate, as sulfuric acid plays critical role in nucleation? In the sentence "Scenario 1: H2SO4 vapor is relatively sufficient against NucOrg, and J8 is therefore determined mainly by the availability of NucOrg vapor. A good correlation is theoretically expected for J8 and NMINP". To be honest, I do not understand the logics behind.*

**Response:** In the revision, we add more clarification to better defense our arguments. We also try our best to explain the difference between our analysis and the reviewer's thoughts.

We assume that NPF rapidly stops after dozens of minutes bursting. This is consistent with huge measurements of banana-shaped NPF events reported in literature (Kulmala et al., 2004). If NPF continuously occurs during the whole event, a fan-shaped NPF event would be detected instead of banana-shaped NPF event. This is because of continuous formation and growth of new particles. In fact, a fan-shaped NPF event was hardly observed. Moreover, in Fig. 1 published by Yue et al, 2010 (Atmos. Environ.) and Fig. 3-4 published by Wang et al., 2011 (Atmos. Chem. Phys.), NPF rapidly stops after dozens of minutes bursting with rapid consumption of $H_2SO_4$ vapor. These studies also directly supports our assumption. We will add the part of analysis in the revision.

The reviewer commented "note that the formation rate is closely related to the increase of nucleation mode particles if looking at the calculation formula of formation rate." The comment does not sound scientific. This is no doubt that FR is determined mainly by the nucleation mechanism (i.e., nucleation of sulfuric acid vapor enhanced by organics). The equation is used to measure the apparent formation rate and has nothing to do with nucleation mechanisms. Technically, we can measure the vehicle speed on basis of vehicle traveling mileage in a fixed time. We clearly know that a vehicle speed depend mainly on engines and fuels, etc., but has nothing to do with vehicle traveling mileage. Moreover, the largest mileage of a vehicle is mainly determined by the used liters of fuel in vehicle tank. Engines and other factors can greatly affect vehicle speed, but the influence on the largest traveling mileage is not comparable to that of the liters of fuel in vehicle tank. The same can be said for FR and NMINP. In NPF events, the authors technically consider sulfuric acid vapor as fuel while organic as engine and other factors affecting vehicle speeds.

The reviewer commented "the authors state that "the NMINP was always determined by the consumed $H_2SO_4$ vapor for nucleation". Sorry for that I cannot accept this view. How about the number of nucleation mode particles when the organic vapors facilitated the nucleation and particle growth to the detectable size?" The authors may disagree with the comments. Regarded much low nucleation rates of inorganic vapors reported so far, the authors strongly believe that all NPF events observed in the atmospheric boundary layer worldwide were facilitated by organic vapors to some extent. When the organic vapors don't facilitate the nucleation and particle growth to the detectable size, there are no NPF events to be observed in the atmospheric boundary layer and the NMIMP is zero.

We also agree that the threshold of formation rate, i.e. 8 $cm^{-3}s^{-1}$, may be coincidentally consistent with our previous study. This needs more work to be confirmed. The part will be

added in the revision.

*10. Page 9, lines 29-33. The concurrent occurrences of class II NPF events at the coastal site and over the marginal seas could not be an evidence of the regional characteristics. The particle number distributions at the two sites were quite different on the days specified by the authors (Figure A1). Besides, it is difficult to convince me with the backward trajectories. The two sites were in a same region under the influence of the same monsoon. Even so, the air masses could be totally different in chemical compositions when they passed over the different cities. With no chemical information or mesoscale simulation, it is hard to say the two sites were interacted and the regional NPF events occurred at the two sites.*

**Response:** Referred to our response to Comment 2, the reviewer may mix up a few concepts. The class II NPF events lasted for 3-5 hours and these NPF events should be considered as regional events.

*11. Page 10, lines 1-6. Condensable vapors are of course critical in NPF. However, it is not reasonable to simply attribute the different characteristics of NPF to the abundances of the condensable vapors. Other factors, such as the preexisting particles and the meteorological conditions also influence the NPF. In this case, more preexisting particles with larger diameters existed at the marginal sea site. Could this also account for the insignificant particle growth?*

**Response:** We agree that weather conditions can affect the growth of newly formed particles by changing gas-aerosol partitioning. We will revise the part accordingly: "Theoretically, higher CS can remove more condensable vapors and consequently reduce the vapor pressure of precursors of various volatilities. In this case, the apparent particle growth was undetectable in the marine atmosphere, with the smaller CS of $0.6\pm0.1$ ($10^{-2}$ $s^{-1}$) versus the value of $5.3\pm0.2$ ($10^{-2}$ $s^{-1}$) in the coastal atmosphere. However, the apparent growth of new particles at the OUC site indicated that 1) the concentrations of condensable vapors were higher than the required value to support the growth and 2) CS was not the dominant factor that determined the growth. In addition to the condensation vapor, weather conditions can affect the growth of newly formed particles by changing gas-aerosol partitioning. "

We may disagree with other parts of reviewer's comments. Preexisting particles can remove condensable vapors, nucleating clusters and newly formed particles from the atmosphere and then affect NPF and the growth of new particles. In addition to affecting condensable vapors, the authors cannot figure out other pathways for preexisting particles to affect particle growing larger than 10 nm.

*12. Section 4.1. I do not agree that new particle formation occurred in this case, i.e. 30 August 2015.*

**Response:** The NPF event on 30 August 2015 followed the definition proposed by Dal Maso et al. (2005), Hirsikko et al. (2007) and Kulmala et al. (2012), i.e., the nucleation mode of newly formed particle was observed for about 6 hours, and newly formed particles grew up to approximately 20 nm. In the revision, the figure of time series of $N_{<30\ nm}$ and CS (show as Fig. R3) will be added and discussed. "Here, we delve into the characteristics and evidence of oceanic precursors that were related to the NPF event on 30 August 2015. The transport pathway on that day is first zoomed in Fig. 6a. As illustrated in Fig. 6b, the NPF event was first observed at 09:40 under meteorological conditions with an ambient temperature of 26°C,

high relative humidity of 74%, and low wind speed of 1.5 m s$^{-1}$ (not shown). During the first hour, the $N_{<30nm}$ increased from $0.6\times10^3$ cm$^{-3}$ to $1.7\times10^3$ cm$^{-3}$. The weaker NPF was associated with higher CS ($2\times10^{-2}$ s$^{-1}$). When the CS decreased to approximately $1\times10^{-2}$ s$^{-1}$ after 11:00, $N_{<30nm}$ sharply increased to $3\times10^3$ cm$^{-3}$, and $D_{pg}$ increased from 13 nm to 18 nm over the following 3 hr with a growth rate of 1.7 nm h$^{-1}$. The signal of new particles disappeared at approximately 16:00. The overall NMINP was 5-20 times lower than all the other NPF events over the marginal seas, and the overall FR of 0.3 cm$^{-3}$ s$^{-1}$ was the minimum in this study. "

[Figure]

Fig. R3 Time series of $N_{<30\,nm}$ and CS on 30 August 2015.

We are sorry for color bar used in Fig. 6b (original version), which may mislead the reviewer. To make the weak signal of new particles to be visible, the scale of color bar in Fig. 6b was one fifth of other contour figures. The choice also makes the signal of pre-existing particles darker in Fig. 6b in comparison with other contour figures.

*13. Page 11, lines 12-15. Figure 7c does not show the altitude variation of the backward trajectories.*

**Response:** We will add the altitudes in the supplementary (add as Fig. S7, show as Fig. R4).

[Figure]

Fig. R4 Height of the air mass back trajectories on 8 and 13 April 2014 over the NWPO.

*14. Page 11, lines 16-26. The sampling periods of MOUDI samples were after the NPF events,*

*not including the hours when the new particles were formed and grew up. I would doubt the reasonability of using these data to infer the chemical species dominating NPF. Same for the other similar discussions.*

**Response:** We thank the comment. In the original version, the inclusion of MOUDI data on those two days are not well justified. The analysis is also too speculative to be convincing. The part will be revised as below:

"One set of MOUDI samples was collected during the period from 11:12 to 23:33. Although the sampling period had a several-hour delay against the NPF period on that day, the air mass back trajectories that swept over the oceanic zone were highly consistent between the two periods (Fig. S8, show as Fig. R5). The concentrations of particulate chemical species were thereby used to argue the polluted extent of the air mass during these periods. The mass concentration of nss-sulfate and oxalate in particles smaller than 10 μm was 1.9 μg m$^{-3}$ and 0.12 μg m$^{-3}$ (derived from Fig. 7d), respectively, higher than on other non-NPF days in this study. Previous studies, e.g., Mukai et al., (1995), Matsumoto et al. (1997), and Jung et al. (2014), reported that the mass concentration of nss-sulfate was approximately 0.5 μg m$^{-3}$ in the clean background over the NWPO. The elevated concentration of nss-sulfate and oxalate on 8 April suggested enhanced anthropogenic precursors input, which was very likely originated from the continent of Japan based on the calculated air mass back trajectories (Fig. 7c, d). The MOUDI data implied that the NPF event likely occurred in the air masses rich in anthropogenic precursors.

Compared to the above event on 8 April, the event on 13 April showed a longer NPF duration, lasting from 07:50 to approximately 08:50 (Fig. 7b). New particles signals were intermittently observed, and the FR was difficult to calculate. The total particle number concentrations increased from 0.3×10$^4$ cm$^{-3}$ to a maximum of 2.6×10$^4$ cm$^{-3}$ during the NPF event, and the NMINP was 1.4×10$^4$ cm$^{-3}$. D$_{pg}$ increased from 8 nm to 14 nm in one hour, and the estimated GR was 3.6 nm h$^{-1}$. One set of MOUDI samples was collected immediately after the event during the period from 09:10 to 21:05. Again, the calculated air mass back trajectories were consistent between the NPF period and the MOUDI sampling period (Fig. S8). The mass concentration of nss-sulfate and oxalate in particles smaller than 10 μm was only 0.6 μg m$^{-3}$ and 0.05 μg m$^{-3}$. These values were close to the clean background of the NWPO, indicating much lower anthropogenic input on 13 April (Fig. 7e). Interestingly, the NMINP values were similar during the two NPF events, although the air mass on 8 April was polluted by anthropogenic inputs. However, the causes of these NPF events require further study because of lack of precursor vapor measurements."

[Figure]

Fig. R5 24-h air mass back trajectory throughout the NPF event and sampling periods (from

7:00 to 24:00 on 8 April and from 7:00 to 21:00 on 13 April)

*15. Page 12, lines 8-10. I do not understand the logics behind this inference, though it is true that the AR increased after Dpg was higher than 50 nm. Why not present the number concentration of >50 nm particles or its fraction in total particles against the NCCN? It would be a more direct way to link the particles larger than 50 nm to CCN.*

**Response:** "threshold" indeed causes misleading. Not all particles larger than 50 nm can be activated as CCN. In the revision, the part has been revised as "At SS of 0.4%, the $D_{pg}$ increased from 19 nm to 50 nm during 10:40-13:10 (black circles in Fig. 8a) with the AR fluctuating at 0.1-0.2 (Fig. 8c). After 13:10, the $D_{pg}$ increased from 50 nm to 77 nm with an increasing AR from ~0.2 to ~0.4. These results are consistent with those in the literature, i.e., particles smaller than 50 nm are unlikely to be activated as CCN at SS=0.4% (Dusek et al., 2006; Petters and Kreidenweis, 2007)."

Following the reviewer's comments, we plotted time series of the number concentration of >50 nm particles ($N_{>50nm}$) and the $N_{CCN}$ in Fig. R6 (blue line). Variations between $N_{>50\ nm}$ and AR are clearly inconsistent. For example, AR showed an increasing trend from 0.2 to 0.4 during 13:10-15:00. $N_{>50\ nm}$ decreased from $1.8 \times 10^4$ cm$^{-3}$ to $1.1 \times 10^4$ cm$^{-3}$ during the period of 13:10-13:30, then increasing to $2.4 \times 10^4$ cm$^{-3}$ at 14:30, followed a decreasing trend after 14:30. Atmospheric particles with the diameter larger than 50 nm include not only the grown new particles, but also preexisting particles. The inconsistency is not very surprised. The reviewer's comment is valid only when the number concentration of preexisting particles >50 nm was either near constant or was negligible relative to grown new particles during the growth period. This is not the fact. Therefore, we disagree with the reviewer on this point.

[Figure]

Fig. R6 Time series of total particle number concentration ($N_{CN}$), number concentration of >50 nm particles ($N_{>50\ nm}$), CCN number concentration at the SS of 0.4% ($N_{CCN(0.4)}$), and CCN activation ratio (AR) on 4 September 2015 in BS.

*16. Caption of Figure 3, what does "exteriors" mean? Why should they be excluded from the regression? Figure 4, what does the black dots represent, same for the other figures? Figure 9, what does the highlighted area denote for?*

**Response:** It should be outlier rather than exterior. We are sorry for our language problem. In Figure 3a, there was a moderately good linear correlation at FRs $\leq$ 8 cm$^{-3}$ s$^{-1}$. The data points with FRs larger than 10 cm$^{-3}$ s$^{-1}$ are deviated largely from the regression curve obtained from the data with FRs $\leq$ 8 cm$^{-3}$ s$^{-1}$ and are thereby treated as outliers. For example, in the linear regression question of [NMINP]=3.9×10$^3$×FR, r=0.83 P<0.01, we consider three times of standard deviation for the slope. At the FR of 11.8 cm$^{-3}$ s$^{-1}$, the range of NMINP is predicted from 3.56×10$^4$ to 5.64×10$^4$ particles cm$^{-3}$. The observed NMINP was only 1.17×10$^4$ particles cm$^{-3}$ and largely deviated from the range. In Figure 3b, the black triangle represents the GR of 26.3 nm h$^{-1}$ and also deviated largely from the regression curve obtained from other data. The point is also treated as an outlier. This will be added in the figure caption.

The black dots in the contour plot of NPF events (Fig. 4, Fig. 6b, Fig. 7a, b, Fig. 8a and Fig. A1) represent the fitted geometric median diameter of new particles ($D_{pg}$) in 1-min time resolution. The clarification will be added in the revised caption of Fig. 4.

In the revision, the shading in Fig. 9 will be removed to avoid any misleading.

*17. The manuscript needs to be grammatically checked by an editing company or a native English speaker professor.*

**Response:** Thanks. The revised version will be language-edited.

**Reference:**

[revised manuscript text omitted]

---

## Author Response (AR2)

*Response to reviewer's comments*

We numbered the comments as 1) and 2) because they are different issues.

1) *The authors have well addressed most of my concerns on the manuscript, though I would still like to ask for clarifications at some points. Firstly, the authors defined that NMINP referred to the net maximum increase in nucleation mode particles number concentration, and FR was calculated according to formula (1) in Text S2 where Ndp was the particle number concentration of nucleation mode particles. Please clarify what are the differences between NMINP and the integration of dNdp/dt over the time period of NPF. I am aware that FR is controlled by many factors but absolutely not the calculation formula. However, it can be expected that higher FR leads to larger enhancement of nucleation mode particles, except that the newly formed particles are removed in some pathways as described in the formula (1) in Text S2. In the cases that the FRs exceeded 8 cm-3 s-1, no correlation existed between NMINP and FRs, could the authors clarify what accounted for the removal of the newly formed particles? The essential influencing factors (FR= kNucOrg[H2SO4]m[NucOrg]n) cannot be used to explain the poor correlation, though which is widely-known to be true, because FR was not calculated according to this formula but based on the observed concentrations of nucleation mode particles.*

**Response:** In the last-round revision, page 5, lines 23-25, we presented "The net maximum increase in the nucleation mode particle number concentration (NMINP) was defined as $N_{<30 nm}$ at the time of reaching the maximal value minus $N_{<30 nm}$ at the time immediately before the apparent NPF was initiated (Zhu et al., 2017)." On basis of the definition, it is clear that NMINP is approximately equal to the integration of dNdp/dt over the initial NPF times. We feel no revision is needed here.

The reviewer argued the statistical analysis results, i.e., "In the cases that the FRs exceeded 8 $cm^{-3}$ $s^{-1}$, no correlation existed between NMINP and FRs" could be due to the removal of the newly formed particles. The possibility is probably low because no significant difference between CS under FR> 8 $cm^{-3}$ $s^{-1}$ and FR≤ 8 $cm^{-3}$ $s^{-1}$. However, the initial NPF times, defined as the time of $N_{<30 nm}$ reaching the maximal value minus the time immediately before the apparent NPF was initiated (Δt), are significantly shorter under FR> 8 $cm^{-3}$ $s^{-1}$ than FR≤ 8 $cm^{-3}$ $s^{-1}$. Note that the dNdp/dt≈NMINP/Δt during the initial NPF period.

Thus, in the revision, we add "Moreover, there was no significant difference between CS under FR> 8 $cm^{-3}$ $s^{-1}$ and FR≤ 8 $cm^{-3}$ $s^{-1}$. The removal of the newly formed particles cannot explain the presence and absence of correlations obtained above. However, the initial NPF times, defined as the time of $N_{<30 nm}$ reaching the maximal value minus the time immediately before the apparent NPF was initiated, are significantly shorter under FR> 8 $cm^{-3}$ $s^{-1}$ than FR≤ 8 $cm^{-3}$ $s^{-1}$. The large FRs, i.e., larger than 8 $cm^{-3}$ $s^{-1}$, are most likely due to the organic-enhanced NPF."

2) *It is very interesting that the authors describe sulfuric acid as the fuel and organics as the engine in NPF. However, I cannot agree the statement that NMINP was always determined by the consumed H2SO4 vapor for nucleation, unless the authors could provide solid*

*evidences to prove that organics are always sufficient and has no/very low possibility to be the limiting factor in NPF. For example, in the cases that the concentrations of the condensable vapors of the organics are much lower than those of the sulfuric acid, could it possible that NMINP is determined by the organic vapors consumed? Overall, these questions are worthy of further discussions before accepting this paper. I look forward to the responses from the authors for clearer clarifications.*

**Response:** We agree "in the cases that the concentrations of the condensable vapors of the organics are much lower than those of the sulfuric acid, NMINP is determined by the organic vapors consumed." The consumed organic vapors then determine the consumed $H_2SO_4$ vapor for nucleation. This is what we said in the last round revision. To be more straight forwards and avoid confusion, the revised sentence reads as "However, the NMINP is always determined by the total consumed $H_2SO_4$ vapor for NPF, although the consumed $H_2SO_4$ was determined by the consumed organic vapor in Scenario 1."